

# 1  Microbial communities associated with sediments and
# 2  polymetallic nodules of the Peru Basin

Massimiliano Molari[1], Felix Janssen[1,2], Tobias Vonnahme[1,a], Frank Wenzhöfer[1,2] and
Antje Boetius[1,2]
[1] Max Planck Institute for Marine Microbiology, Bremen, Germany
[2] HGF-MPG Joint Research Group on Deep Sea Ecology and Technology, Alfred
Wegener Institute for Polar and Marine Research, Bremerhaven, Germany
[a] Present address: UiT the Arctic University of Tromsø, Tromsø, Norway
*Correspondence to*: Massimiliano Molari (mamolari@mpi-bremen.de)





**Abstract.** Industrial-scale mining of deep-sea polymetallic nodules will need to remove nodules in
large areas of the seafloor. The regrowth of the nodules by metal precipitation is estimated to take
millions of years. Thus for future mining impact studies, it is crucial to understand the role of nodules
in shaping microbial diversity and function in deep-sea environments. Here we investigated microbial
community composition based on 16S rRNA gene sequences retrieved from sediments and nodules of
the Peru Basin (>4100 m water depth). The nodule field of the Peru Basin showed a typical deep-sea
microbiome, with dominance of the classes Gammaproteobacteria, Alphaproteobacteria,
Deltaproteobacteria, and Acidimicrobiia. Nodules and sediments host distinct bacterial and archaeal
communities, with nodules showing lower diversity and a higher proportion of sequences related to
potential metal-cycling bacteria (i.e. Magnetospiraceae, Hyphomicrobiaceae), bacterial and archaeal
nitrifiers (i.e. *AqS1*, unclassified Nitrosomonadaceae, *Nitrosopumilus*, *Nitrospina*, *Nitrospira*), and
bacterial sequences found in ocean crust, nodules, hydrothermal deposits and sessile fauna. Sediment
and nodule communities overall shared a low proportion of Operational Taxonomic Units (OTU; 21 %
for Bacteria and 19 % for Archaea). Our results show that nodules represent a specific ecological niche
(i.e. hard substrate, high metal concentrations and sessile fauna), with a potentially relevant role in
organic carbon degradation. Differences in nodule community composition (e.g. Mn-cycling bacteria,
nitrifiers) between the Clarion-Clipperton Fracture Zone (CCZ) and the Peru Basin suggest that
changes in environmental setting (i.e. sedimentation rates) play also a significant role in structuring the
nodule microbiome.



## 1 Introduction

Polymetallic nodules (or manganese nodules) occur in abyssal plains (4000−6000 m water depth) and consist primarily of manganese and iron, as well as many other metals and rare earth elements (Crerar and Barnes, 1974; Kuhn et al. 2017). Nodules are potato- or cauliflower-shaped structures with typical diameters of 4−20 cm and are typically found at the sediment surface or occasionally buried in the uppermost 10 cm sediment horizon. The mechanisms of nodule formation are not completely elucidated. The current understanding is that they are formed via mineral precipitations from bottom waters (*Hydrogenetic* growth) or pore waters (*Diagenetic* growth) involving both abiotic and microbiological processes  (Crerar and Barnes, 1974; Riemann, 1983; Halbach et al., 1988; Wang et al., 2009). The formation of nodules is a slow process that is estimated to range between thousands and millions of years per millimetre growth (Kerr, 1984; Boltenkov, 2012).

Rising global demand for metals has renewed interests in commercial mining of deep-sea nodule deposits. Mining operations would remove nodules, disturb or erode the top decimeters of sediment, and create near bottom sediment plumes that will resettle and cover the seafloor (Miller et al., 2018). Although the first nodules have been discovered in the 1870's (Murray, 1891), only little is known about the biodiversity, biological processes and ecological functions of the nodules and their surrounding sediments as specific deep-see habitat. Major questions remain, for example as to spatial turnover on local and global scales, the role of the microbial community in and around nodules, the role of nodules as substrate for endemic species. Hence, there is the need to thoroughly characterize baseline conditions as a requirement for any mining operations as these will require assessments of impacts associated with mining.

Extensive and dense nodule fields are found in different areas of the Pacific and Indian deep seas. Nodule accumulations of economic interest have been found in four geographical locations: the Clarion-Clipperton Fracture Zone (CCZ) and the Penrhyn Basin in the central north and south Pacific Ocean, respectively; the Peru Basin in the south-east Pacific; and in the center of the north Indian Ocean (Miller et al., 2018). Previous work on the structure of microbial communities of nodule fields by 16S rRNA gene sequencing focused on the CCZ and the central South Pacific Ocean (Xu et al., 2007; Wu et al., 2013; Tully and Heidelberg, 2013; Blöthe et al., 2015; Shulse et al., 2016; Lindh et al., 2017). All studies showed that polymetallic nodules harbor microorganisms that are distinct from the surrounding sediments and overlying water. They indicate that nodule communities show a pronounced spatial variability, but these results are so far not conclusive. Similar microbial communities were observed in nodules collected at distances of 6000 km and 30 km (Wu et al., 2013; Shulse et al. 2016), while Tully and Heidelberg (2013) found that nodule communities varied among sampling sites (<50 km). Besides, potential Mn-oxidizers and -reducers such as *Alteromonas*, *Pseudoalteromonas*, *Shewanella* and *Colwellia* were proposed as a core of the nodule microbiome involved in the formation of nodules (Wu et al. 2013; Blöthe et al., 2015), but they were not found in all nodules sampled so far (Tully and Heidelberg, 2013; Shulse et al. 2016). The lack of knowledge on the diversity and composition of microbial assemblages of other nodule provinces makes it difficult to assess whether observed differences within the CCZ may reflect regional differences in environmental conditions (e.g.

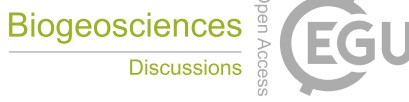

input of organic matter, bathymetry, topography, sediment type), or in abundance and morphology of
nodules, or in the colonization of the nodules by epifauna and protozoans.
In this study we investigated the diversity and composition of bacterial and archaeal communities
associated with manganese nodule fields of the Peru Basin. The Peru Basin is located about 3000 km
off the coast of Peru and covers about half of the size of the CCZ, which is 5.000–9.000 km away. The
present-day organic carbon flux in this area is approximately two times higher than in the CCZ,
resulting in higher content of organic carbon in the surface sediments (>1% *vs* 02−06% in the CCZ),
and a shallower oxic-suboxic front (10 cm *vs* tens of meters sediments depth in the CCZ; Müller et al.,
1988; Heakel et al., 2001; Volz et al., 2018). As a consequence of differences in environmental
conditions (e.g. organic carbon flux, carbonate compensation depth, sediment type, topography and
near-bottom currents), the Peru basin and the CCZ host manganese nodules with different geological
features (Kuhn et al. 2017). This includes: i) nodules from the Peru Basin are often larger, with a
typical cauliflower shape, compared to those in CCZ that have a discoidal shape and a size of 2-8 cm
(Kuhn et al. 2017); ii) average nodule abundance in the Peru Basin is lower (10 kg m$^{-2}$) than in CCZ
(15 kg m$^{-2}$; Kuhn et al. 2017); iii) Mn nodules from the Peru Basin are thought to be mainly formed by
suboxic diagenesis, whereas CCZ nodules apparently exhibit a mixture of diagenetic and hydrogenetic
origin (von Stackelberg 1997; Chester and Jickells 2012); iv) while Peru Basin and CCZ nodules
consist of the same type of mineral (disordered phyllomanganates), they have a different metal content
(Wegorzewski and Kuhn 2014; Wegorzewski et al. 2015).
An increasing number of studies and policy discussions address the scientific basis of ecological
monitoring in deep-sea mining, highlighting the need to identify appropriate indicators and standards
for environmental impact assessments and ecological management. A key aspect is avoiding harmful
effects to the marine environment, which will have to include loss of species and ecosystem functions.
The aims of this study was to assess the structure and similarity of benthic microbial communities of
nodules and sediments of the Peru Basin nodule province, and to compare them with those of other
global deep-sea sediments and nodules in the CCZ. The focus was on similarity comparisons in order
to investigate endemism and potential functional taxa that could be lost due to the removal of
manganese nodules by mining activities. To achieve this, the hypervariable 16S rRNA regions V3-V4
for Bacteria and, V3-V5 for Archaea were amplified from DNA extracted from nodules and
surrounding sediments and sequenced using the Illumina paired-end MiSeq platform. The hypotheses
tested were i) nodules shape deep-sea microbial diversity and functions, ii) nodules host specific
microbial community compared to the surrounding sediments, and iii) environmental setting and
nodule features impact microbial community composition.



## 2 Methods

### 2.1 Sample collection

Sediment samples and polymetallic nodules were collected as a part of the MiningImpact project of the Joint Programming Initiative JPI Healthy and Productive Seas and Oceans (JPIOcean) on board of R/V Sonne (expedition SO242/2; 28[th] of August - 1[st] of October 2015) in the Peru Basin around 7° S and 88.5° W. Samples were collected at three sites outside the seafloor area selected in 1989 for a long-term disturbance and recolonization experiment (DISCOL; Thiel et al., 2001), for this reason they were called "References Sites". Sediment samples were collected using TV-guided MUltiple Corer (TV-MUC) at three stations per site (Table 1). The cores were sliced on board in a temperature-controlled room (set at *in situ* temperature), and aliquots of sediment were stored at −20 °C for DNA extraction and prepared for cell counts (see sections below). Manganese nodules where sampled, using a TV-MUC, or a Remotely Operated Vehicle (ROV Kiel6000). The nodules where partly located at the surface or buried down to 3 cm below the seafloor (bsf) with diameters of a few cm. Nodules were gently rinsed with 0.22-µm filtered cold bottom seawater to remove adhering sediment, stored in sterile plastic bags at −20 °C and crushed before DNA extraction in the home lab.

### 2.2 DNA extraction and sequencing

The nodules collected with the MUC where crushed and stored at −20 °C. From the nodules collected with the ROV, only the surface layer was scraped off using a sterile spoon, and subsequently crushed and frozen (−20 °C). The DNA was extracted from 1 g of wet sediment (0-1 cm layer) and from 1 g of wet nodule's fragments using the FastDNA[TM] SPIN Kit for Soil (Q-BIOgene, Heidelberg, Germany) following the manual protocol. An isopropanol precipitation was performed on the extracted DNA, and DNA samples were stored at −20 °C. As control for DNA contamination (negative control), DNA extraction was carried out on purified water after being in contact with sterile scalpel and plastic bag.

Amplicon sequencing was done at the CeBiTec laboratory (Centrum für Biotechnologie, Universität Bielefeld) on an Illumina MiSeq machine. For the 16S amplicon library preparation we used the bacterial primers 341F (5′-CCTACGGGNGGCWGCAG-3′) and 785R (5′-GACTACHVGGGTATC TAATCC-3′), and the archaeal primers Arch349F (5′-GYGCASCAGKCGMGAAW-3′) and Arch915R (5′-GTGCTCCCCCGCCAATTCCT-3′) (Wang and Qian, 2009; Klindworth et al., 2013), which amplify the 16S rDNA hypervariable region V3-V4 in Bacteria (400−425 bp fragment length) and the V3-V5 region in Archaea (510 bp fragment length). The amplicon library was sequenced with the MiSeq v3 chemistry, in a 2x300 bp paired run with >50,000 reads per sample, following the standard instructions of the 16S Metagenomic Sequencing Library Preparation protocol (Illumina, Inc., San Diego, CA, USA).

The quality cleaning of the sequences was performed with several software tools. CUTADAPT (Martin, 2011) was used for primer clipping. Subsequently the TRIMMOMATIC software (Bolger et al., 2014) was used to remove low-quality sequences (for Bacteria SLIDINGWINDOW:4:10 MINLEN:300; for Archaea SLIDINGWINDOW:6:13 MINLEN:450): In case of bacteria data this step was performed before the merging of reverse and forward reads with PEAR (Zhang et al., 2014) while merging of the archaeal data set was done after removing low-quality sequences in order to enhance the





number of retained reads for long archaeal 16S fragments. Clustering of sequences into OTUs
(operational taxonomic units) was done using the SWARM algorithm (Mahé et al., 2014). The
taxonomic classification was based on the SILVA rRNA reference database (release 132), at a
minimum alignment similarity of 0.9, and a last common ancestor consensus of 0.7 (Pruesse et al.,
2012). Raw sequences with removed primer sequences were deposited at the European Nucleotide
Archive (ENA) under accession number PRJEB30517 and PRJEB32680; the sequences were archived
using the service of the German Federation for Biological Data (GFBio; Diepenbroek et al., 2014).

The total number of sequences obtained in this study is reported in table 2. Absolute singletons
(SSOabs), i.e. OTUs consisting of sequences occurring only once in the full dataset (Gobet et al., 2013)
were removed (Table 2). Similarly, contaminant sequences (as observed in the negative control) and
unspecific sequences (i.e., bacterial sequences in the archaeal amplicon dataset, and archaeal,
chloroplast, and mitochondrial sequences in the bacterial dataset) were removed from amplicon data
sets before the analysis (Table 2). The dominant OTU sequences and OTU sequences highly abundant
in the nodules were subjected to BLAST search (BLASTn; Gene Bank nucleotide database
12/06/2019) in order to identify in which others habitats the closest related (i.e. >99 %) sequences have
been previously reported.

**2.3 Data analysis**

The first three Hill Numbers, or the effective number of species, were used to describe alpha-diversity:
species richness ($H_0$), the exponential of Shannon entropy ($H_1$), and the inverse Simpson index ($H_2$;
Chao et al., 2014). Coverage-based and sample-size-based rarefaction (based on actual number of
sequences) and extrapolation (based on double number of sequences) curves were calculated for the
Hill's numbers using the R package iNEXT (Hsieh et al., 2018). Calculation of the estimated richness
(Chao1) and the identification of unique OTUs (present exclusively in one sample) were based on
repeated (n = 100) random subsampling of the amplicon data sets. Significant differences in alpha-
diversity indices between substrates (i.e. manganese nodules and sediments) were determined by
analysis of variance (ANOVA), or by non-parametric Kruskal-Wallis test (KW) when ANOVA's
assumptions were not satisfied.

Beta-diversity in samples from different substrates and from the substrate in samples from different
sites was quantified by calculating Bray-Curtis dissimilarity based on centred log-ratio (CLR)
transformed OTU abundances and Jaccard dissimilarity based on a presence/absence OTU table. The
latter was calculated with 100 sequence re-samplings per sample on the smallest dataset (40613
sequences for Bacteria and 1835 sequences for Archaea). Bray-Curtis dissimilarity was used to produce
non-metric multidimensional scaling (NMDS) plots, and the Jaccard dissimilarity coefficient was used
to calculate the number of shared OTUs between samples. The permutational multivariate analysis of
variance (PERMANOVA; Anderson, 2001) was used to test difference in community structure and
composition.

Differentially abundant OTUs and genera were detected using the R package ALDEx2 (Fernandes et
al. 2014) at a significance threshold of 0.01 and 0.05 for Benjamini–Hochberg (BH) adjusted
parametric and non-parametric (KW) P-values, respectively. We only discussed the taxa that were at



least two times more abundant in nodules than in sediments (i.e. (Log2(Nodule/sediment) ≥1) and with
a sequences contribution of total number of sequences ≥1 % (for genera) or ≥0.1 % (for OTUs).
All statistical analyses were conducted in R using the core distribution with the additional packages
vegan (Oksanen et al. 2015), compositions (Van den Boogaart et al., 2014), iNEXT (Hsieh et al.,
2018), and ALDEx2 (Fernandes et al. 2014).
**3 Results**
**3.1 Microbial alpha-diversity**
Bacterial and archaeal communities in 5 nodules and 9 sediment samples (Table 1) were investigated
using specific sets of primers for Bacteria and Archaea on the same extracted pool of DNA per station.
The number of bacterial sequences retrieved from DNA extracted from sediments and nodules was on
average 5±5 and 25±14 times higher, respectively, than those obtained for archaea (t-test: p<0.001,
df=11, t=4.5).
Table 2 shows the statistics of sequence abundance and proportion of singletons and cosmopolitan
types. Sequence abundances of bacteria were comparable between sediments and nodules.
Cosmopolitan OTU; i.e. those present in 80 % of the sediments and nodule samples were only 9 % of
all taxa (77 % of all sequences), whereas rare OTU occurring only in <20 % of all samples represented
50 % the taxa (4 % of all sequences). Sediments *vs* nodules contained only 4 and 2 %, respectively, of
endemic taxa, defined as those were abundant in either substrate but rare in the other. Thus the
contribution of unique OTUs to the total number of OTUs was lower in manganese nodules than in
sediments samples (Table 3, Figure 1a). Bacterial and archaeal diversity was investigated calculating
the total number of OTUs (Hill number q=0; $H_0$) and the estimated richness (Chao1), and the unique
OTUs (present exclusively in one station). For this analysis, the latter were calculated with sequence
re-sampling, to overcome differences in sequencing depth. Abundance-based coverage estimators,
exponential Shannon (Hill number q=1; $H_1$) and inverse Simpson (Hill number q=2; $H_2$), were also
calculated. The rarefaction curve indicates that the richness ($H_0$) of the less abundant and rare OTUs
was somewhat underestimated both in nodules and in sediments (Figure S1 a-b). However, the
bacterial and archaeal diversity was well described for the abundant OTUs ($H_1$ and $H_2$; Figure S1 a-b);
with more than 90 % of the estimated diversity covered (Figure S1 c-d). Both in sediments and nodules
the alpha-diversity indices were higher for Bacteria than for Archaea (t-test: p<0.0001, df=12,
t=8.0−16.0), while the contribution of unique OTUs to the total number of OTUs was comparable
(Table 3). Bacterial communities in manganese nodules have lower Hill numbers and Chao1 indices
compared to those associated to sediments (Table 3, Figure 1a). Archaeal communities showed the
same patterns for diversity indices and unique OTUs, with exception for $H_2$ index that did not show
significant difference between nodules and sediments (Table 3, Figure 1b).
**3.2 Patterns in microbial community composition**
The changes in microbial community structure at OTU level (beta-diversity) between substrates and
samples were quantified by calculating Bray-Curtis dissimilarities from CLR transformed OTU





abundance. Shared OTUs were estimated by calculating Jaccard dissimilarity from OTU
presence/absence based on repeated random subsampling of the amplicon data sets. Microbial
communities associated with manganese nodules differed significantly from those found in the
sediments (Figure 2, Table S1). Also, significant differences were detected in sediment microbial
community structure among the different sites (PERMANOVA; Bacteria: $R^2 = 0.384$; $p = 0.003$; $F_{2,8} =$
1.87; Archaea: $R^2 = 0.480$; $p = 0.013$; $F_{2,8} = 2.31$; Table S1), and between communities associated with
nodules and sediment at Reference South (PERMANOVA; Bacteria: $R^2 = 0.341$; $p = 0.023$; $F_{1,6} = 2.59$;
Archaea: $R^2 = 0.601$; $p = 0.029$; $F_{1,6} = 7.53$; Table S1). "Site" defined by geographic location, and
"Substrate", i.e. origin from sediment or nodule, explained a similar proportion of variation in bacterial
community structure (27 % and 23 %, respectively). "Substrate" had a more important role in shaping
archaeal communities than "Site" (explained variance 35 % and 19 %, respectively; Table S1). The
number of shared OTUs between nodules and sediments (Bacteria: 21 %; Archaea: 19 %) was lower
than those shared within nodules (Bacteria: 30 %; Archaea: 30 %) and within sediments (Bacteria: 31
%; Archaea: 32 %) (Figure 4).
Bacterial communities in manganese nodules and sediments were dominated by the classes
Gammaproteobacteria (26 %), Aphaproteobacteria (19 %), Deltaproteobacteria (9 %), Bacteroidia (5
%), Acidimicrobiia (4 %), Dehalococcoidia (4 %), Planctomycetacia (4 %), Nitrospinia (3 %), and
Phycisphaerae (3 %), which accounted for more than 75 % of the total sequences (Figure 3). All
archaeal communities were dominated by Thaumarchaeota (*Nitrosopumilales*), which represented more
than 95% of all sequences. The remaining small proportion of sequences was taxonomically assigned
to *Woesearchaeia* (Figure 4b). Nodule and sediment samples showed similar compositions of most
abundant bacterial genera (contribution to total number of sequence ≥1 %; Figure 4a). 69 bacterial
genera (9 % of all genera) were differentially abundant in the nodules and in the sediment, accounting
for 36 % and 21 % of total sequences retrieved from nodules and sediments, respectively (ALDEx2:
ANOVA adjusted $p<0.01$ and KW adjusted $p<0.05$; Table 4). Of those only one unclassified genus
within the family of Sphyngomonadaceae and genus *Filomicrobium* were exclusively found in nodules
and not in the sediment samples, and their contribution to the total number of sequences was less than
0.06 %. Genera that were more abundant in the nodules than in the sediments included: unclassified
Alphaproteobacteria (7 %), *Nitrospina* (4 %), unclassified SAR324 clade (Marine group B; 3 %),
unclassified Hyphomicrobiaceae (3 %), Pirellulaceae Pir4 lineage (2 %), unclassified
Methyloligellaceae (1 %), unclassified Pirellulaceae (1 %), *Acidobacteria*, unclassified Subgroup 9 (1
%) and Subgroup 17 (1 %), Nitrosococcaceae *AqS1* (1 %), Calditrichaceae *JdFR-76* (1 %), and
*Cohaesibacter* (1 %) (Table 4). In the sediment we identified 21 genera that were more abundant than
in the nodules, but all together they represented only 3 % of total sequences recovered from sediments.
128 OTUs were highly abundant in nodules (ALDEx2: ANOVA adjusted $p<0.01$ and KW adjusted
$p<0.05$), which accounts for 24 % of total sequences retrieved from nodules (Table 5a). The closest
related sequences (≥99 % similarity) were retrieved from ocean crusts (30 %), from nodule fields (26
%), from hydrothermal/seep sediments and deposits (21 %), from worldwide deep-sea sediments (16
%), and associated to invertebrates (7 %; Table 5b and Figure 5).



## 4 Discussion

Industrial-scale mining of deep-sea polymetallic nodules may remove nodules and the active surface seafloor layer at a spatial scale ranging from ca. 50,000−75,000 km$^2$ per claim to ca. 1 million km$^2$ including all current exploration licences (Miller et al., 2018). The regrowth of nodules will take millions of years, thus it is unknown if the associated biota could recover at all (Simon-Lledo et al., 2019). The response of microbial communities to the loss of nodules and seafloor integrity is largely unknown. It may play an important role in the ecological state of the seafloor habitat due to the many functions bacteria and archaea hold in the food-web, element recycling, and biotic interactions, beyond representing the largest biomass in deep-sea sediments (Joergensen and Boetius 2007). It is thus crucial to understand the role of nodules in shaping microbial diversity and in hosting microbes with important ecological functions. So far, only few studies were carried out to investigate the microbiota of nodule fields, and most of them were focused on identifying microbes involved in metal cycling. Here, we investigated similarity of microbial community structures in sediments and nodules retrieved from the Peru Basin. The objectives of this study were: i) to compare the microbes of nodules fields with microbiota of deep-sea sediments, in order to identify specific features of microbial diversity of nodule fields; ii) to elucidate differences in diversity and in microbial community structure between sediments and nodules, and their relapses on potential microbially-mediated functions; iii) to understand the major drivers in shaping microbial communities associated to the nodules.

### 4.1 Microbial diversity of nodule fields is distinct from other deep-sea areas

Benthic bacterial assemblages in sediments and nodules of the Peru basin showed typical dominance of the classes Gammaproteobacteria, Aphaproteobacteria, Deltaproteobacteria, and Acidimicrobiia, as reported for worldwide deep-sea sediments (Bienhold et al., 2016; Fig. 4) and in the Pacific Nodule Province (Wang et al., 2010; Wu et al., 2013; Shulse et al., 2016; Lindh et al., 2017). But at higher taxonomic resolution we detected substantial differences to the microbial community composition of other deep-sea regions.  Sediments of the Peru Basin bacteria classes were depleted in sequence abundances of Flavobacteria, Gemmetimonadetes and Bacilli, whereas sequence abundances of the Chloroflexi (i.e. Dehalococcoidia), Planctomycetes (i.e. Pirellulaceceae, Phycisphaeraceae) and the genus *Nitrospina* were higher compared to other deep-sea regions (Bienhold et al., 2016, Varliero et al., 2019).  Dehalococcoidia and Planctomycetes were previously reported as important component of benthic microbial assemblages in the Pacific Ocean (Wang et al., 2010; Wu et al., 2013; Blöthe et al., 2015; Walsh et al., 2016; Lindh et al., 2017). Their contribution to the total community was found to increase in organic matter depleted subsurface sediments (Durbin and Teske, 2011; Walsh et al., 2016).

Dominant OTUs (>1 %) belonged to unclassified Actinomarinales, Gammaproteobacteria, Subgroup 21 (phylum Acidobacteria), and to genus *Woesia* (family Woeseiaceae). Members of Actinomarinales and Woeseiaceae are cosmopolitan types in deep-sea sediments (Bienhold et al., 2016). For Actinomarinales there are no cultivates, and the function of this group remains unknown. In the case of Woeseiaceae, one representative is in culture (*Woeseia ocaeni*). *W. ocaeni* is an obligate chemoorganoheterotroph (Du et al., 2016), suggesting a role in organic carbon remineralization for



members of that family, as recently confirmed by analysis of deep-sea assembled genomes (Hoffmann
et al. in revision). Closest related sequences of Subgroup 21 have been reported in deep-sea sediments
(Schauer et al. 2010) and across Pacific nodule fields (Wu et al., 2013), but also in association with
deep-sea benthic giant foraminifera (Xenophyophores) and in surrounding sediments (Hori et al.,
2013). The subgroup 21-like OTU was also one of the 10 most abundant OTUs retrieved from nodules
(0.9 %). Xenophyophores have agglutinated tests and can grow to decimetre size, suggesting that
members of Subgroup 21 may be colonists of biological and/or hard substrates.
Within the class Alphaproteobacteria the most abundant OTUs (>0.5 %) belonged to unclassified
genera of the families Magnetospiraceae (order Rhodospirillales), Hyphomicrobiaceae (order
Rhizobiales), and Kiloniellaceae (order Rhodovibrionales). Magnetospiraceae and Hyphomicrobiaceae
are the most abundant families in nodules with >2 % of OTUs. Closely related sequences have been
reported previously across Pacific Nodule Provinces (Xu et al., 2007; Shulse et al., 2016). The family
of Magnetospiraceae includes microaerophilic heterotrophs, able of magnetotaxis and iron reduction
(i.e. genus Magnetospirillum; Matsunaga et al. 1991; Schuler and Frankel, 1999), and thus the
members of this family could play a role in Fe(III) mobilization, affecting its bioavailability.
Hyphomicrobiaceae-like sequences found in this study are related to genera *Hyphomicrobium* and
*Pedomicrobium* (sequence identity 97 %), which have been reported to be involved in manganese
cycling (Tyler, 1970; Larsen et al., 1999; Stein et al., 2001). A potential contribution of these groups in
metal cycling in manganese nodules is also suggested by the presence of closest related sequences in
ocean crust (Santelli et al., 2008; Lee et al., 2015), which typically hosts epilithic and endolithic
microbial communities of chemolithotrophic metals-oxidizers (Staudigel et al., 2008). Similarly,
Kiloniellaceae related OTUs might be involved in metal-cycling as closely related sequences were
found in marine basalts (Mason et al., 2007; Santelli et al., 2008) and inside other manganese nodules
(Blöthe et al., 2015). Most of the marine cultivates in the family Kiloniellaceae belong to genus
*Kiloniella*, that have been isolated from marine macroalga (Wiese et al., 2009), the guts of Pacific
white shrimp (Wang et al., 2015), marine sponge (Yang et al., 2015), spider crab and clam (Gerpe et
al., 2017), and from the surface water of a polynia in the Western Antarctic Sea (Si et al., 2017).
Besides, Kiloniellaceae-like sequences were found in sponges (Cleary et al., 2013), sea start larvae
(Galac et al., 2016) and in seamount's iron mats (Scott et al., 2017). The presence of rich sessile and
mobile metazoan communities associated to nodules offers various potential hosts for members of
Kiloniellaceae. *Kiloniella* is a chemoheterotrophic aerobe, and the draft genome of an isolate from the
gut of a Pacific white shrimp shows potential for denitrification and iron acquisition and metabolism
(Wang et al., 2015). Thus, either as free-living or host-associated life, the potential contribution of
Kiloniellaceae in metal cycling requires further investigation.
Archaea were also present in sediments of the Peru Basin, with Nitrosopumilaceae (phylum
Thaumarchaeota) dominating the archaeal communities (Figure 4b). In contrast to what was reported
for CCZ (Tully and Heidelberg, 2013; Shulse et al., 2016), Archaeal sequences comprised a lower
portion of total sequences retrieved from sediments and nodules of Peru basin (ca. 10 %), and they



were lower in nodules compared to the sediments. The majority of member of Nitrosopumilaceae are
believed to be capable of oxidation of ammonia to nitrite, the first step of nitrification (Offre et al.,
2013). Archaeal ammonia oxidizers have a higher affinity for ammonia than bacterial ammonia
oxidizers, and they are favoured in environments with low ammonia concentrations (Martens-Habbena
et al., 2009). The Peru Basin has higher particulate organic-carbon fluxes as compared to central
Pacific Ocean (Haeckel et al., 2001; Mewes et al., 2014), which results in higher remineralisation rates
and higher ammonia fluxes. These limit the thickness of oxygenated sediments to 10 cm in the Peru
Basin while they can reach up to 2-3 m depth in the CCZ (Haeckel et al., 2001; Mewes et al., 2014;
Volz et al., 2018). Hence differences observed between CCZ and Peru nodule fields in the contribution
of archaeal sequences to microbial assemblages are likely due to ammonia availability, which is
controlled by organic matter fluxes.

**4.2 Microbial community structure differs between sediments and nodules**

Beta-diversity of microbial community structure in the Peru sediments showed remarkable OTU
turnover already on a local scale (<60 km; Figure 4), which is at the higher end for turnover rates from
previous microbial beta-diversity estimates for bathyal and abyssal seafloor assemblages (Jacob et al.,
2013; Ruff et al., 2015; Bienhold et al., 2016; Walsh et al., 2016; Varliero et al., 2019). Here we
focused specifically on the contribution of nodules to diversity, which could be a critical parameter in
the ecological assessment of nodule removal. Analysis of community composition at OTU level shows
that nodules and sediments host distinct bacterial and archaeal communities (Figure 2). Albeit the
microbial communities in the sediment showed significant differences between sites, the low number
of shared OTUs between sediments and nodules <20% supports the presence of specific bacterial and
archaeal communities associated with polymetallic nodule habitats (Wu et al., 2013; Tully and
Heidelberg, 2013; Shulse et al., 2016; Lindh et al. 2017). However, the proportion of truly endemic,
unique nodule OTUs was also low (Table 3, Figure 1a), nonetheless it is relevant to highlight that
nodule removal would lead to a loss of specific types of microbes in a mined deep-sea region (Blöthe et
al., 2015).
Microbial communities associated to nodules are generally less diverse than those in the sediments, and
the decrease in diversity was observed both in rare and abundant bacterial types (Figure 1 and S1). This
seems to be a common feature of polymetallic nodules (Wu et al., 2013; Tully and Heidelberg, 2013;
Zhang et al., 2014; Shulse et al., 2016; Lindh et al. 2017). Tully and Heidelberg (2013) suggested that
it might be due to less availability of potential energy sources (e.g. organic matter) compared to
sediments. Despite that the sedimentation rate exceeds the growth rate of nodules, the nodules are
typically exposed to bottom water and not covered by sediments (Peukert et al., 2018). Although, it is
unknown whether physical mechanisms (e.g. current regime or seasonal events) or biological processes
(e.g. grazing, active cleaning) are responsible for lack of sediments accumulation on nodules, the
decrease of microbial diversity with the decrease of organic matter availability is in accordance with
positive energy-diversity relationship reported for deep-sea sediments (Bienhold et al., 2012).
However, the presence of foraminiferal assemblages (Gooday et al., 2015) and specific sessile



metazoan communities (Vanreusel et al., 2016) on the surface of nodules may represent a potential
source of transformed organic matter (e.g. dissolved organic matter) and catabolic products, which may
represent a much more valuable energy source for microbes than refractory particulate organic matter
sinking from water column. Furthermore, higher microbial diversity in the sediments than in the
nodules could be the result of the accumulation of allochthonous microbes, as suggested by the higher
proportion of rare and unique OTUs in the sediments. Lastly, the nodules offer hard substrate and
presence of metals, which can select for specific Bacteria and Archaea. Similarly, hydrothermal
deposits have typically lower bacterial diversity than deep-sea sediments despite chemical energy
sources being highly available (Ruff et al., 2015; Wang et al., 2018). We propose that the decreased
diversity of abundant OTUs in nodules, observed especially for Bacteria, suggests selection for
colonists adapted to specific ecological niches associated with nodules (e.g. high metals concentration,
hard substrate, presence of sessile fauna).
**4.3 Potential functions of microbial communities associated to nodules**
The presence of a large proportion of bacterial community with low abundance in the sediments, but
enriched by the nodule environment at the level of genera (35 %) and OTUs (24 %) (Table 4 and 5a)
indicates niche specialization. The most abundant OTUs (13 % of bacterial community) in nodules
include unclassified Hyphomicrobiaceae, Magnetospiraceae, Alphaproteobacteria, Arenicellaceae and
SAR324, *Nitrospina*, *AqS1*, Methyloligellaceae, Subgroup 9, Subgroup 17, Kiloniellaceae,
*Cohaesibacter* and JdFR-76, which closest related sequences have been retrieved from Pacific nodules
(e.g. Wu et al. 2013; Blöthe et al., 2015), basaltic rocks (e.g. Mason et al 2007; Santelli et al 2008;
Mason et al., 2009; Lee et al., 2015), sulfide and carbonate hydrothermal deposits (e.g. Sylvan et al.,
2012; Kato et al., 2015), and giant foraminifera (Hori et al., 2013; Table 5b and Figure 5). There are
currently no cultivated representatives and metabolic information for these members of the Bacteria,
and it is not known whether they have metal tolerance mechanisms or they are actively involved in
metal cycling. The high abundance of potential metal reducers (i.e. Magnetospiraceae) and oxidizers
(i.e. Hyphomicrobiaceae), and presence of encrusting protozoans (Gooday et al., 2015), microbial
eukaryotes (Shulse et al., 2016) and metazoans (Vanreusel et al., 2016) create specific ecological
niches, which may be at least partially responsible for the observed selection of microbial taxa in
nodules. Overall, these findings suggest that bacterial groups adapted to lithic or biological substrates
preferentially colonize nodules, likely favoured by manganese and iron availability, formation of
biofilms and presence of sessile fauna communities.
The reduction and dissolution of Mn oxides by dissolved organic matter (e.g. humic compounds)
occurs typically in photic or reducing aquatic environments (Sunda et al., 1983; Stone and Morgan,
1984; Stone, 1987; Sunda and Huntsman, 1994). However reductively dissolution of Mn oxides by
dissolved organic substrates has been observed also in dark oxygenated seawater (Sunda et al., 1983;
Sunda and Huntsman, 1994), suggesting that it could be a relevant abiotic process in manganese
nodules. Indeed, this reaction yields manganese(II) and low-molecular-weight organic compounds
(Sunda and Kieber, 1994), which potentially may favour Mn-oxidizing Bacteria and microbial
exploitation of refractory dissolved organic matter. Intense extracellular enzymatic activities have been





reported for seafloor-exposed basalts (Meyers et al., 2014), raising the question of whether the close
related microbes associated to nodules might have comparable degradation rates. Furthermore, nodules
host diversified communities of suspension feeders such as serpulid tubeworms, sponges, corals and
crinoids (Vanreusel et al., 2016), which filter microbes and POC from the bottom water and release
DOM and catabolic metabolites (e.g. ammonia). Thus, nodules may act as hot spots of organic carbon
degradation. Albeit metabolic activity has never been quantified on nodules and sequence abundances
are lower, the increased abundance of nitrifiers in nodules compared to the sediments reported for
Pacific Nodule Province (Tully and Heidelberg, 2013; Shulse et al., 2016) and in this study could
indicate a high activity. Nitrifiers catalyse the oxidation of ammonia, a catabolic product of
heterotrophic metabolism, to nitrite and eventually to nitrate. In the CCZ the nitrifier community was
composed of archaeal ammonia-oxidizing *Nitrosopumilus*, which represented a large portion of the
microbial assemblages (up to 20 %), and a minor contribution of bacterial nitrite-oxidizing *Nitrospira*
(Tully and Heidelberg, 2013; Shulse et al., 2016). Peru Basin sediments and nodules showed more
diversified nitrifier communities, which are enriched by ammonia oxidizing *AqS1* (1 %) and
unclassified Nitrosomonadaceae (1 %) and by nitrite-oxidizing *Nitrospina* (4 %) and *Nitrospira* (1 %;
Table 4 and 5). *Nitrospina* are not commonly reported for deep-sea sediments, but they are the
dominant nitrite oxidizers in the oceans (Luecker et al., 2013). They have recently been reported as
symbiont of deep-sea glass sponges (Tian et al., 2016), which also commonly colonize FeMn nodules
(Vanreusel et al., 2016). The Nitrospina-related OTUs detected in the nodules showed only low
similarity with pelagic *Nitrospina gracilis* and *Nitrospina*-like sequences found in deep-sea glass
sponge (sequence identity of 93 %), but were closely related with sequences recovered from marine
basalts (Mason et al., 2007; Santelli et al. 2008; Mason et al. 2009), suggesting nodules as a native
habitat.
**5 Conclusions**
The sediments of nodule fields in the Peru Basin host a specific microbial community composition of
bacterial taxa reported for organic carbon poor environments (i.e. Chloroflexi, Planctomycetes) and
potentially involved in metal-cycling (i.e. Magnetospiraceae, Hyphomicrobiaceae). Nodule
communities were distinct from sediments and showed a higher proportion of sequences from potential
Mn-cycling bacteria including bacterial taxa found in ocean crust, nodules and hydrothermal deposits.
Our results are in general agreement with previous studies in the CCZ, confirming that nodules provide
a specific ecological niche. However remarkable differences in community composition (e.g. Mn-
cycling bacteria, nitrifiers) between the CCZ and the Peru Basin in microbial community composition
also show that environmental setting (i.e. POC flux) and features of FeMn nodules (e.g. metal content)
may play a significant role in structuring the nodule microbiome. This indicates that microbial
community structure and function would be impacted by nodule removal, and that regional differences
would need to be assessed, to determine the spatial turnover and the impact on endemic types.
Furthermore, our results suggest that the removal of nodules, and potentially also the blanketing of
nodules with plume sediments suspended during the mining operations may affect the cycling of metal
and other elements. Future work is needed to characterize metabolic activities on and in nodules, and to



understand factors and processes controlling nodule colonization. Specifically, restoration experiments
should take place to test whether artificial substrates favour the recovery of microbial and fauna
communities, and their related ecological functions.
**Data availability**
Raw sequences with removed primer sequences were deposited at the European Nucleotide Archive
(ENA) under accession number PRJEB30517 and PRJEB32680.
**Author contributions**
**A.B**, **F.J**, **F.W.** and **M.M** conceived the study. **A.B**, **F.J**, **F.W.** and **T.W.** performed sampling
activities. **M.M.** compiled and analysed the data. **M.M.** wrote the paper with the contribution from all
Authors.
**Competing interest**
The authors declare that they have no conflict of interest.
**Special issue statement**
**Acknowledgements**
The Authors want to thank captain and crew of the SO242/1 and /2 expedition, and M. Alisch, J.
Bäger, J. Barz, A. Nordhausen, F. Schramm, R. Stiens and W. Stiens, (HGF-MPG Joint Research
Group on Deep Sea Ecology and Technology) for technical support. The Authors are also grateful to
Dr. H. Tegetmeyer for support with Illumina sequencing (CeBiTec laboratory; HGF-MPG Joint
Research Group on Deep Sea Ecology and Technology).
This work was funded by the German Ministry of Research and Education (BMBF grant no.
03F0707A-G) as part of the MiningImpact project of the Joint Programming Initiative of Healthy and
Productive Seas and Oceans (JPIOceans). We acknowledge further financial support from the
Helmholtz Association (Alfred Wegener Institute Helmholtz Center for Polar and Marine Research,
Bremerhaven) and the Max Planck Society (MPG), as well as from the ERC Advanced Investigator
Grant ABYSS (294757) to AB for technology and sequencing. The research has also received funding
from the European Union Seventh Framework Program (FP7/2007- 2013) under the MIDAS project,
grant agreement number 603418.

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

**Figure captions**
Figure 1. Comparison of diversity indices and unique OTUs between manganese nodules and
sediments for (a) bacterial and (b) archaeal communities. $H_0$: number of OTUs (q=0); $H_1$: exponential



Shannon (q=1); $H_2$: inverse Simpson (q=2); Unique: OTUs present exclusively in each station
(percentage relative to total OTUs of whole dataset). Chao1 and Unique OTUs were calculated with
100 sequence re-samplings per sample to the smallest dataset (40613 sequences for Bacteria and 1835
sequences for Archaea). Red line shows the median. F: statistic *F-ratio*, with subscript numbers
reporting the degrees of freedom between groups and within groups, respectively; p: probability level;
KW-test: Kruskal-Wallis test; $\chi^2$: Chi square test value, with subscript numbers reporting the degrees
of freedom between groups and sample size, respectively.
Figure 2. Non-metric multidimensional scaling (NMDS) plot based on Euclidean distance similarity
matrix of bacterial (a) and archaeal (b) community structure at OTU level. Sequence abundances of
OTUs were centre log-ratio transformed. Permutational multivariate analysis of variance
(PERMANOVA) showed significant differences between nodule and sediment associated microbial
communities (for details see Table S1). Each sample (dot) is connected to the weighted averaged mean
of the within group distances. Ellipses represent one SD of the weighted averaged mean.
Figure 3. Bacterial community structure at dominant class level (cut-off ≥1 %). MN: manganese
nodules; MUC: sediments.
Figure 4. Bacterial (a) and Archaeal (b) dominant genera (cut-off ≥ 1 %) for surface nodules and
sediments. Cluster on top of barplot showed dissimilarity in OTUs composition, as defined by Jaccard
dissimilarity index based on presence/absence OTU table and calculated with 100 sequence re-
samplings per sample on the smallest dataset (40613 sequences for Bacteria and 1835 sequences for
Archaea). un.: unclassified. * due to extremely low number of sequences (n=182), this sample was not
included in analysis requiring sequence re-samplings. MN: manganese nodules; MUC: sediments.
Figure 5. Habitats coverage for the closest related sequences (≥99 % similarity) to OTUs highly
abundant in the nodules. For details see Table 4a-b.
**Table captions**
Table 1. Stations list and description of investigated sites/substrates.
Table 2. Statistics of sequence and OTUs abundance, and proportion of absolute singletons,
cosmopolitans and endemics for sediments (n=9) and nodule (n=5 for Bacteria, n=4 for Archaea)
samples collected in Peru Basin. Absolute singletons: OTUs consisting of sequences occurring only
once in the entire dataset; Cosmopolitan: OUTs present in 80 % of sediments and 80 % of nodule
samples; Endemics: OTUs exclusively present only in 80 % sediments (and <20 % of nodule samples)
or in 80 % nodule samples (and <20 % of sediments samples).
Table 3. Bacterial and archaeal diversity indices and unique OTUs for all nodules and sediment
samples. Indices and unique OTUs were calculated without singletons.





Table 4. Genera differentially abundant in nodules and sediments (ALDEx2: glm adjusted p<0.01; KW
adjusted p<0.05). In bold the most abundant genera (≥1 %) at least two times more abundant in nodule
than in sediment; in italic the genera exclusively present (i.e. unique) in nodules. Base 2 logarithm of
the ratios between geometric mean centred sequences number of nodule (Nod) and sediment (Sed), and
average of the sequences contribution of total number of sequences (%) retrieved in nodules and in
sediments are shown.
Table 5. (a) OTUs highly abundant in nodules (ALDEx2: glm adjusted p<0.01; KW adjusted p<0.05).
Only OUTs ≥0.1 % are reported. Base 2 logarithm of the ratios between geometric mean centred
sequences number of nodule (Nod) and sediment (Sed), and average of the sequences contribution of
total number of sequences (%) retrieved in nodules and in sediments are shown. (b) Closest related
sequences as indemnified with BLASTn (NCBI nucleotide database 12/06/2019).



**Figure 1.**

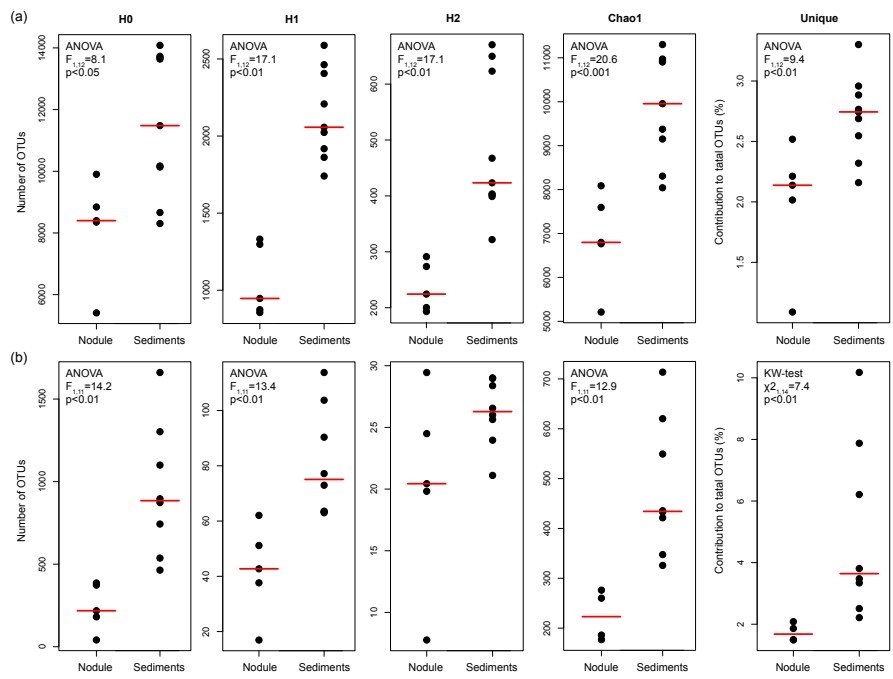




**Figure 2.**

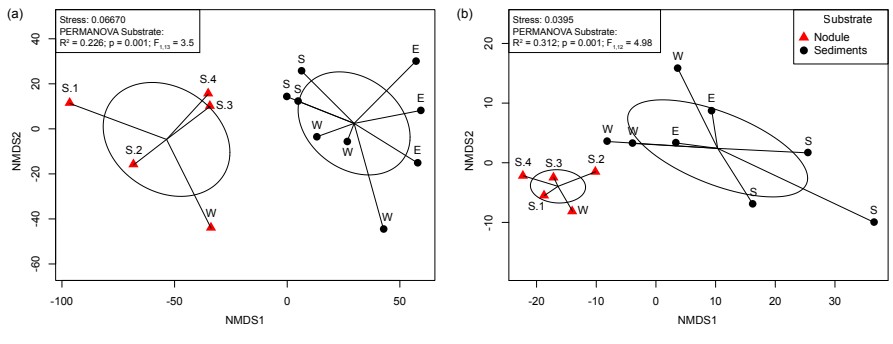




**Figure 3.**

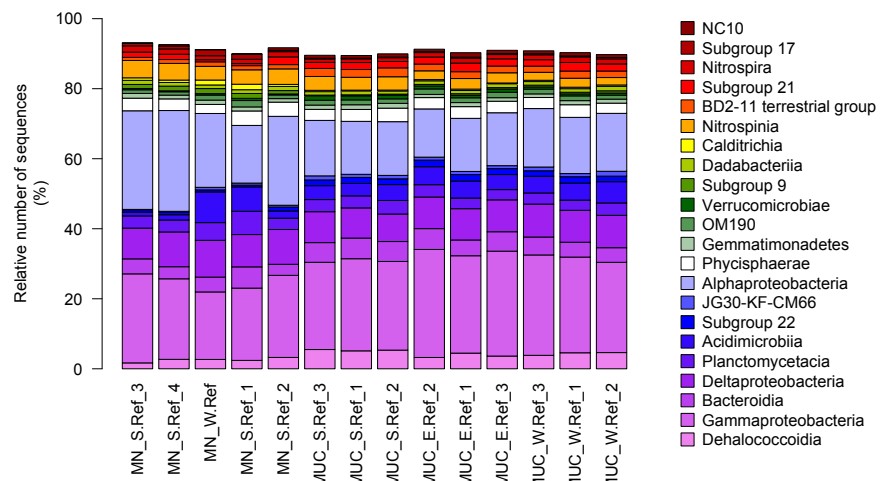




**Figure 4.**

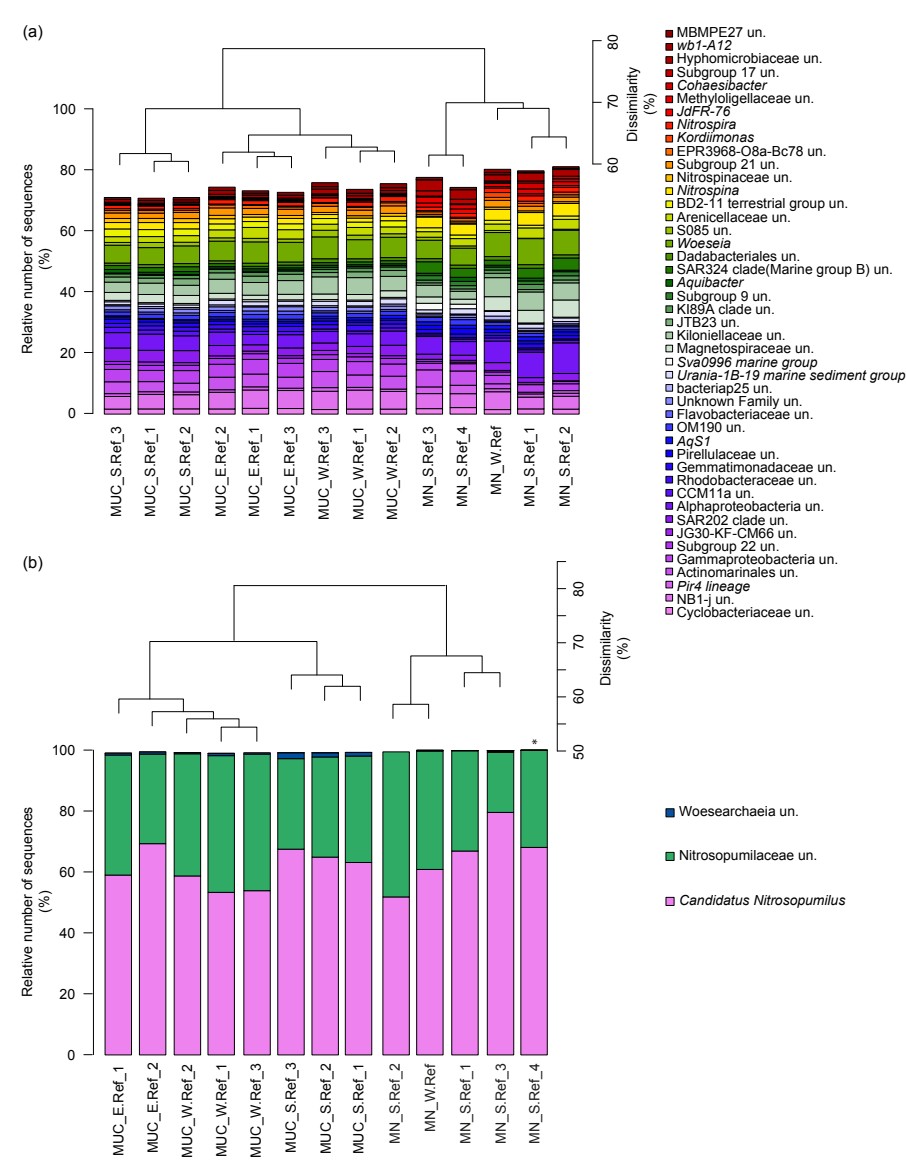





**Figure 5.**

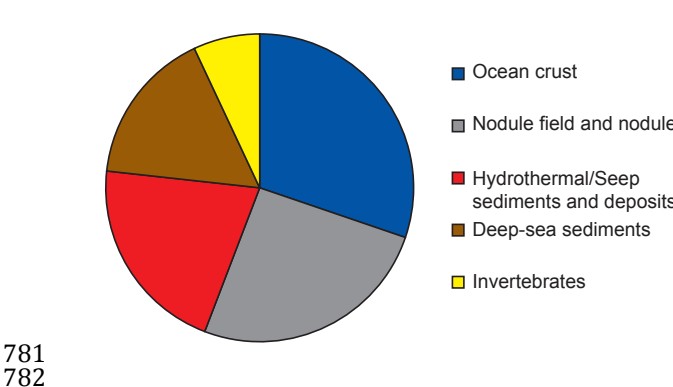




**Table 1.**

| Station | Sample ID | Sampling Time | Latitude (N) | Longitude (E) | Depth (m) | Device | Site | Sediment layer (cm bsf) | Substrate |
|---|---|---|---|---|---|---|---|---|---|
| SO242/2_147 | MUC_E.Ref_1 | 02.09.15 | −7.1007 | −88.414 | 4198.2 | MUC | Reference East | 0-1 | sediments |
| SO242/2_148 | MUC_E.Ref_2 | 02.09.15 | −7.1006 | −88.414 | 4195.8 | MUC | Reference East | 0-1 | sediments |
| SO242/2_151 | MUC_E.Ref_3 | 03.09.15 | −7.1006 | −88.414 | 4197.8 | MUC | Reference East | 0-1 | sediments |
| SO242/2_194 | MN_W.Ref | 15.09.15 | −7.0761 | −88.526 | 4129.5 | MUC | Reference West | surface | nodule |
| SO242/2_194 | MUC_W.Ref_1 | 15.09.15 | −7.0761 | −88.526 | 4129.5 | MUC | Reference West | 0-1 | sediments |
| SO242/2_194 | MUC_W.Ref_2 | 15.09.15 | −7.0761 | −88.526 | 4129.5 | MUC | Reference West | 0-1 | sediments |
| SO242/2_194 | MUC_W.Ref_3 | 15.09.15 | −7.0761 | −88.526 | 4129.5 | MUC | Reference West | 0-1 | sediments |
| SO242/2_198 | MN_S.Ref_1 | 16.09.15 | −7.1262 | −88.450 | 4145.6 | ROV | Reference South | surface | nodule |
| SO242/2_198 | MN_S.Ref_2 | 16.09.15 | −7.1262 | −88.450 | 4145.6 | ROV | Reference South | surface | nodule |
| SO242-2_208 | MN_S.Ref_3 | 19.09.15 | −7.1256 | −88.450 | 4150.7 | MUC | Reference South | surface | nodule |
| SO242-2_208 | MN_S.Ref_4 | 19.09.15 | −7.1256 | −88.450 | 4150.7 | MUC | Reference South | surface | nodule |
| SO242/2_208 | MUC_S.Ref_1 | 15.09.15 | −7.0761 | −88.526 | 4129.5 | MUC | Reference South | 0-1 | sediments |
| SO242/2_208 | MUC_S.Ref_2 | 15.09.15 | −7.0761 | −88.526 | 4129.5 | MUC | Reference South | 0-1 | sediments |
| SO242/2_208 | MUC_S.Ref_3 | 15.09.15 | −7.0761 | −88.526 | 4129.5 | MUC | Reference South | 0-1 | sediments |

MUC: TV-guided MUltiple Corer; ROV: Remote Operated Vehicle (Kiel 6000); bsf: below seafloor.

**Table 2.**

| Bacteria | OTUs n. | OTUs % | Sequences n. | Sequences % |
|---|---|---|---|---|
| Entire dataset | 557468 | | 2271610 | |
| Contaminants | 20 | 0.0 | 15710 | 0.7 |
| Absolute singletons | 525169 | 94.2 (56 / 39) [b] | 525169 | 23.1 (14 / 9) [b] |
| Working dataset [a] | 32279 | 5.8 | 1730731 | 76.2 |
| Sediments dataset [a] | 28666 | 5.1 (88.8) [c] | 1032246 | 45.4 (59.6) [c] |
| Nodule dataset [a] | 19279 | 3.5 (59.7) [c] | 698485 | 30.8 (40.3) [c] |
| Cosmopolitan OTUs [a] | 1452 | 0.5 (8.9) [c] | 1167668 | 58..4 (76.7) [c] |
| Endemics OTUs sediments [a] | 1356 | 0.2 (4.2) [c] | 39895 | 1.8 (2.3) [c] |
| Endemics OTUs nodules [a] | 599 | 0.1 (1.9) [c] | 52328 | 2.3 (3.0) [c] |

| Archaea | OTUs n. | OTUs % | Sequences n. | Sequences % |
|---|---|---|---|---|
| Entire dataset | 51856 | | 293098 | |
| Contaminants | 0 | 0.0 | 0 | 0.0 |
| Absolute singletons | 49482 | 95.4 (77 / 19) [b] | 49482 | 16.9 (14 / 3) [b] |
| Working dataset [a] | 2372 | 4.6 | 243616 | 83.1 |
| Sediments dataset [a] | 2356 | 4.5 (99.3) [c] | 219460 | 74.9 (90.1) [c] |
| Nodule dataset [a] | 591 | 1.1 (24.9) [c] | 24156 | 8.2 (9.9) [c] |
| Cosmopolitan OTUs [a] | 112 | 0.2 (4.7) [c] | 194736 | 66.4 (79.9) [c] |
| Endemics OTUs sediments [a] | 198 | 0.4 (8.3) [c] | 10610 | 3.6 (4.4) [c] |
| Endemics OTUs nodules [a] | 5 | 0.01 (0.2) [c] | 121 | 0.04 (0.05) [c] |

[a] after removal of contaminants (defined by negative control) and absolute singletons sequences (see Materials and Methods for
details), percentage calculated on Entire dataset.
[b] contribution of sediments and nodules to absolute singletons, respectively.
[c] percentage calculated on working dataset.

**Table 3.**





| Bacteria | Sequences n. [a] | Sequences n. [b] | $H_0$ | $H_1$ | $H_2$ | Chao1 [c] | sd | Unique (%) [c] | sd |
|---|---|---|---|---|---|---|---|---|---|
| MUC_E.Ref_2 | 226078 | 161443 | 13638 | 2024 | 402 | 10930 | 201.6 | 2.7 | 0.1 |
| MUC_E.Ref_3 | 218324 | 166847 | 13680 | 2057 | 423 | 10972 | 200.1 | 2.9 | 0.1 |
| MUC_E.Ref_1 | 222924 | 164985 | 14082 | 2208 | 467 | 11302 | 166 | 3.0 | 0.1 |
| MN_W.Ref | 209563 | 159724 | 9902 | 1296 | 290 | 8085 | 143.6 | 2.2 | 0.1 |
| MUC_W.Ref_1 | 137990 | 104301 | 11480 | 1918 | 403 | 9955 | 164 | 2.8 | 0.1 |
| MUC_W.Ref_2 | 112259 | 81103 | 10171 | 1862 | 399 | 9151 | 148.3 | 2.3 | 0.1 |
| MUC_W.Ref_3 | 236896 | 178985 | 13727 | 1741 | 322 | 10905 | 198.9 | 3.3 | 0.1 |
| MN_S.Ref_1 | 313418 | 236498 | 8841 | 853 | 192 | 6798 | 138.3 | 2.5 | 0.1 |
| MN_S.Ref_2 | 220364 | 172668 | 8399 | 872 | 199 | 6766 | 132.8 | 2.1 | 0.1 |
| MN1_S.Ref_3 | 114074 | 43932 | 5409 | 945 | 223 | 5211 | 73.28 | 1.1 | 0.1 |
| MN2_S.Ref_4 | 64218 | 76729 | 8351 | 1329 | 272 | 7594 | 124.2 | 2.0 | 0.1 |
| MUC_S.Ref_1 | 77424 | 65890 | 10137 | 2588 | 670 | 9374 | 110.5 | 2.7 | 0.1 |
| MUC_S.Ref_2 | 58575 | 45832 | 8662 | 2406 | 623 | 8306 | 93.85 | 2.2 | 0.1 |
| MUC_S.Ref_3 | 59503 | 40613 | 8306 | 2463 | 650 | 8041 | 84.73 | 2.5 | 0.1 |


| Archaea | Sequences n. [d] | Sequences n. [b] | $H_0$ | $H_1$ | $H_2$ | Chao1 [c] | sd | Unique (%) [c] | sd |
|---|---|---|---|---|---|---|---|---|---|
| MUC_E.Ref_2 | 40952 | 34494 | 896 | 63 | 21 | 433 | 52 | 3.5 | 0.5 |
| MUC_E.Ref_3 | 25090 | 20215 | 743 | 73 | 26 | 421 | 50 | 3.3 | 0.6 |
| MUC_E.Ref_1 | na | na | na | na | na | na | na | na | na |
| MN_W.Ref | 11737 | 12623 | 373 | 51 | 24 | 260 | 33 | 2.1 | 0.4 |
| MUC_W.Ref_1 | 18097 | 14878 | 537 | 63 | 24 | 348 | 36 | 2.5 | 0.5 |
| MUC_W.Ref_2 | 37656 | 31192 | 873 | 77 | 28 | 436 | 53 | 3.8 | 0.6 |
| MUC_W.Ref_3 | 13031 | 10444 | 464 | 64 | 26 | 326 | 31 | 2.2 | 0.4 |
| MN_S.Ref_1 | 7423 | 5384 | 218 | 38 | 20 | 186 | 26 | 1.5 | 0.3 |
| MN_S.Ref_2 | 15314 | 9472 | 386 | 62 | 29 | 276 | 29 | 1.5 | 0.4 |
| MN1_S.Ref_3 | 6099 | 1835 | 181 | 43 | 20 | 177 | 15 | 1.9 | 0.3 |
| MN2_S.Ref_4 [e] | 722 | 182 | 41 | 17 | 8 | na | na | na | na |
| MUC_S.Ref_1 | 34166 | 29221 | 1100 | 90 | 27 | 549 | 66 | 6.2 | 0.6 |
| MUC_S.Ref_2 | 41633 | 36378 | 1302 | 104 | 29 | 620 | 71 | 7.9 | 0.8 |
| MUC_S.Ref_3 | 75344 | 64433 | 1661 | 114 | 29 | 714 | 75 | 10.2 | 0.9 |


$H_0$: number of OTUs; $H_1$: exponential Shannon; $H_2$: inverse Simpson; Unique: OTUs present exclusively in one station (percentage relative to total OTUs of whole dataset); na: not available.
[a] after the merging of forward and reverse reads;
[b] after removal of un-specific and contaminants sequences (see Materials and Methods for details);
[c] calculated with 100 sequence re-samplings per sample to the smallest dataset (40613 sequences for Bacteria and 1835 sequences for Archaea), average data and standard deviation (sd) are given;
[d] after quality trimming of merged forward and reverse reads;
[e] due to extremely low number of sequences, this sample was not included in analyses requiring sequences re-sampling.





**Table 4.**

| Enriched in Nodule | LOG2(Nod/Sed) | Nodule (%) | Sediment (%) | Enriched in Sediment | LOG2(Nod/Sed) | Nodule (%) | Sediment (%) |
|---|---|---|---|---|---|---|---|
| *Sphingomonadaceae_unclassified* | – | *0.04* | *0.00* | Planctomycetales_unclassified | -0.02 | 0.44 | 0.49 |
| *Filomicrobium* | – | *0.01* | *0.00* | Lutibacter | -1 | 0.00 | 0.02 |
| Geminicoccaceae_unclassified | 4 | 0.12 | 0.01 | Chloroflexi_unclassified | -2 | 0.03 | 0.09 |
| Methyloceanibacter | 4 | 0.17 | 0.02 | AT-s3-28_unclassified | -2 | 0.03 | 0.09 |
| Robiginitomaculum | 4 | 0.09 | 0.00 | Chitinophagales_unclassified | -2 | 0.05 | 0.16 |
| Mesorhizobium | 3 | 0.25 | 0.01 | Bacteriovoracaceae_unclassified | -2 | 0.04 | 0.14 |
| **Cohaesibacter** | **3** | **0.78** | **0.10** | Nannocystaceae_unclassified | -2 | 0.02 | 0.07 |
| OPB56_unclassified | 3 | 0.03 | 0.00 | Cellvibrionaceae_unclassified | -2 | 0.02 | 0.08 |
| 67-14_unclassified | 3 | 0.31 | 0.06 | OM182 clade_unclassified | -2 | 0.13 | 0.47 |
| Syntrophaceae_unclassified | 3 | 0.06 | 0.01 | Candidatus Komeilibacteria_unclassified | -2 | 0.01 | 0.03 |
| Maribacter | 3 | 0.06 | 0.01 | Roseobacter clade NAC11-7 lineage | -2 | 0.04 | 0.11 |
| **Methyloligellaceae_unclassified** | **2** | **1.46** | **0.31** | Bacteroidia_unclassified | -2 | 0.03 | 0.11 |
| Entotheonellaceae_unclassified | 2 | 0.20 | 0.04 | IS-44 | -2 | 0.05 | 0.20 |
| Blastocatella | 2 | 0.18 | 0.04 | Oligoflexaceae_unclassified | -2 | 0.01 | 0.08 |
| Calorithrix | 2 | 0.03 | 0.01 | Lentimicrobiaceae_unclassified | -2 | 0.01 | 0.04 |
| **Hyphomicrobiaceae_unclassified** | **2** | **2.72** | **0.71** | Marinoscillum | -3 | 0.02 | 0.08 |
| Planctomicrobium | 2 | 0.05 | 0.01 | Anaerolineaceae_unclassified | -3 | 0.05 | 0.36 |
| Simkaniaceae_unclassified | 2 | 0.13 | 0.03 | Colwelliaceae_unclassified | -3 | 0.01 | 0.13 |
| Microtrichaceae_unclassified | 2 | 0.13 | 0.03 | Subgroup 7_unclassified | -3 | 0.00 | 0.03 |
| LD1-PA32_unclassified | 2 | 0.05 | 0.01 | Peredibacter | -3 | 0.01 | 0.05 |
| **Subgroup 17_unclassified** | **2** | **1.03** | **0.27** | Marinimicrobia (SAR406 clade)_unclassified | -4 | 0.01 | 0.07 |
| **JdFR-76** | **2** | **0.93** | **0.26** | Total | | 1.00 | 2.91 |
| **Subgroup 9_unclassified** | **2** | **1.26** | **0.42** | | | | |
| Chlamydiales_unclassified | 2 | 0.16 | 0.06 | | | | |
| **SAR324 clade(Marine group B)_unclassified** | **2** | **3.12** | **1.10** | | | | |
| Vermiphilaceae_unclassified | 2 | 0.14 | 0.05 | | | | |
| Acanthopleuribacter | 1 | 0.04 | 0.01 | | | | |
| Bythopirellula | 1 | 0.04 | 0.01 | | | | |
| **Nitrospina** | **1** | **3.79** | **1.72** | | | | |
| Gemmataceae_unclassified | 1 | 0.04 | 0.02 | | | | |
| Planctomycetacia_unclassified | 1 | 0.05 | 0.03 | | | | |
| SM1A02 | 1 | 0.29 | 0.15 | | | | |
| Ekhidna | 1 | 0.17 | 0.09 | | | | |
| Phycisphaeraceae_unclassified | 1 | 0.49 | 0.27 | | | | |
| **AqS1** | **1** | **1.10** | **0.66** | | | | |
| Microtrichales_unclassified | 1 | 0.19 | 0.08 | | | | |
| **Pirellulaceae_unclassified** | **1** | **1.31** | **0.75** | | | | |
| pItb-vmat-80_unclassified | 1 | 0.05 | 0.00 | | | | |
| **Pir4 lineage** | **1** | **1.60** | **0.91** | | | | |
| **Alphaproteobacteria_unclassified** | **1** | **7.15** | **4.44** | | | | |
| Babeliales_unclassified | 1 | 0.10 | 0.07 | | | | |
| Parvularculaceae_unclassified | 1 | 0.06 | 0.04 | | | | |
| PAUC43f marine benthic group_unclassified | 1 | 0.54 | 0.36 | | | | |
| Subgroup 10 | 0 | 0.75 | 0.67 | | | | |
| Aquibacter | 0 | 0.83 | 0.73 | | | | |
| Cyclobacteriaceae_unclassified | 0 | 1.76 | 1.70 | | | | |
| Gemmatimonadaceae_unclassified | 0 | 1.17 | 1.15 | | | | |
| Rhodothermaceae_unclassified | 0 | 0.38 | 0.39 | | | | |
| Total | | 35.37 | 17.82 | | | | |




**Table 5.**
(a)

| Phylum | Class | Order | Family | Genus | OTU | LOG2 (Nod/Sed) | Nodule (%) | Sediment (%) |
|---|---|---|---|---|---|---|---|---|
| Proteobacteria | Alphaproteobacteria | Rhizobiales | Hyphomicrobiaceae | Hyphomicrobiaceae_unclassified | otu29 | 2 | 2.5 | 0.7 |
| Proteobacteria | Alphaproteobacteria | Rhodospirillales | Magnetospiraceae | Magnetospiraceae_unclassified | otu11 | 2 | 2.2 | 0.7 |
| Proteobacteria | Alphaproteobacteria | Alphaproteobacteria_unclassified | Alphaproteobacteria_unclassified | Alphaproteobacteria_unclassified | otu31 | 8 | 1.5 | 0.0 |
| Proteobacteria | Alphaproteobacteria | Alphaproteobacteria_unclassified | Alphaproteobacteria_unclassified | Alphaproteobacteria_unclassified | otu83 | 8 | 0.9 | 0.0 |
| Proteobacteria | Alphaproteobacteria | Alphaproteobacteria_unclassified | Alphaproteobacteria_unclassified | Alphaproteobacteria_unclassified | otu160 | 6 | 0.3 | 0.0 |
| Proteobacteria | Alphaproteobacteria | Alphaproteobacteria_unclassified | Alphaproteobacteria_unclassified | Alphaproteobacteria_unclassified | otu249 | 7 | 0.2 | 0.0 |
| Proteobacteria | Deltaproteobacteria | SAR324 clade(Marine group B) | SAR324 clade(Marine group B)_unclassified | SAR324 clade(Marine group B)_unclassified | otu66 | 8 | 0.7 | 0.0 |
| Proteobacteria | Deltaproteobacteria | SAR324 clade(Marine group B) | SAR324 clade(Marine group B)_unclassified | SAR324 clade(Marine group B)_unclassified | otu78 | 2 | 0.6 | 0.1 |
| Proteobacteria | Deltaproteobacteria | SAR324 clade(Marine group B) | SAR324 clade(Marine group B)_unclassified | SAR324 clade(Marine group B)_unclassified | otu202 | 3 | 0.4 | 0.0 |
| Proteobacteria | Deltaproteobacteria | SAR324 clade(Marine group B) | SAR324 clade(Marine group B)_unclassified | SAR324 clade(Marine group B)_unclassified | otu317 | 1 | 0.2 | 0.1 |
| Proteobacteria | Deltaproteobacteria | SAR324 clade(Marine group B) | SAR324 clade(Marine group B)_unclassified | SAR324 clade(Marine group B)_unclassified | otu947 | 4 | 0.2 | 0.0 |
| Proteobacteria | Deltaproteobacteria | SAR324 clade(Marine group B) | SAR324 clade(Marine group B)_unclassified | SAR324 clade(Marine group B)_unclassified | otu588 | 2 | 0.1 | 0.0 |
| Proteobacteria | Deltaproteobacteria | SAR324 clade(Marine group B) | SAR324 clade(Marine group B)_unclassified | SAR324 clade(Marine group B)_unclassified | otu425 | 2 | 0.1 | 0.0 |
| Nitrospinae | Nitrospinia | Nitrospinales | Nitrospinaceae | Nitrospina | otu68 | 3 | 1.4 | 0.1 |
| Nitrospinae | Nitrospinia | Nitrospinales | Nitrospinaceae | Nitrospina | otu227 | 6 | 0.2 | 0.0 |
| Nitrospirae | Nitrospinia | Nitrospinales | Nitrospinaceae | Nitrospira | otu215 | 3 | 0.2 | 0.0 |
| Nitrospirae | Nitrospinia | Nitrospinales | Nitrospinaceae | Nitrospira | otu636 | 6 | 0.2 | 0.0 |
| Nitrospinae | Nitrospinia | Nitrospinales | Nitrospinaceae | Nitrospira | otu434 | 3 | 0.1 | 0.0 |
| Proteobacteria | Gammaproteobacteria | Arenicellales | Arenicellaceae | Arenicellaceae_unclassified | otu36 | 2 | 1.4 | 0.4 |
| Proteobacteria | Gammaproteobacteria | Arenicellales | Arenicellaceae | Arenicellaceae_unclassified | otu162 | 5 | 0.4 | 0.0 |
| Proteobacteria | Gammaproteobacteria | Steroidobacterales | Woeseiaceae | Woeseia | otu97 | 4 | 0.5 | 0.0 |
| Proteobacteria | Gammaproteobacteria | Steroidobacterales | Woeseiaceae | Woeseia | otu266 | 2 | 0.2 | 0.1 |
| Proteobacteria | Gammaproteobacteria | Steroidobacterales | Woeseiaceae | Woeseia | otu521 | 6 | 0.2 | 0.0 |
| Proteobacteria | Gammaproteobacteria | Steroidobacterales | Woeseiaceae | Woeseia | otu346 | 4 | 0.2 | 0.0 |
| Proteobacteria | Gammaproteobacteria | Steroidobacterales | Woeseiaceae | Woeseia | otu991 | 5 | 0.1 | 0.0 |
| Proteobacteria | Alphaproteobacteria | Rhizobiales | Methyloligellaceae | Methyloligellaceae_unclassified | otu113 | 2 | 0.5 | 0.1 |
| Proteobacteria | Alphaproteobacteria | Rhizobiales | Methyloligellaceae | Methyloligellaceae_unclassified | otu184 | 7 | 0.3 | 0.0 |
| Proteobacteria | Alphaproteobacteria | Rhizobiales | Methyloligellaceae | Methyloligellaceae_unclassified | otu234 | 3 | 0.2 | 0.0 |
| Acidobacteria | Subgroup 9 | Subgroup 9_unclassified | Subgroup 9_unclassified | Subgroup 9_unclassified | otu255 | 6 | 0.5 | 0.0 |
| Proteobacteria | Gammaproteobacteria | Nitrosococcales | Nitrosococcaceae | AqS1 | otu122 | 2 | 0.8 | 0.3 |
| Acidobacteria | Subgroup 17 | Subgroup 17_unclassified | Subgroup 17_unclassified | Subgroup 17_unclassified | otu326 | 5 | 0.7 | 0.0 |
| Acidobacteria | Subgroup 17 | Subgroup 17_unclassified | Subgroup 17_unclassified | Subgroup 17_unclassified | otu865 | 1 | 0.1 | 0.1 |
| Calditrichaeota | Calditrichia | Calditrichales | Calditrichaceae | JdFR-76 | otu171 | 1 | 0.6 | 0.2 |
| Calditrichaeota | Calditrichia | Calditrichales | Calditrichaceae | JdFR-76 | otu541 | 4 | 0.2 | 0.0 |
| Proteobacteria | Alphaproteobacteria | Rhodovibrionales | Kiloniellaceae | Kiloniellaceae_unclassified | otu357 | 3 | 0.1 | 0.0 |
| Proteobacteria | Alphaproteobacteria | Rhodovibrionales | Kiloniellaceae | Kiloniellaceae_unclassified | otu435 | 4 | 0.1 | 0.0 |
| Proteobacteria | Alphaproteobacteria | Rhodovibrionales | Kiloniellaceae | Kiloniellaceae_unclassified | otu370 | 3 | 0.1 | 0.0 |
| Proteobacteria | Alphaproteobacteria | Rhodovibrionales | Kiloniellaceae | Kiloniellaceae_unclassified | otu467 | 6 | 0.1 | 0.0 |
| Proteobacteria | Alphaproteobacteria | Rhodovibrionales | Kiloniellaceae | Kiloniellaceae_unclassified | otu450 | 2 | 0.1 | 0.0 |
| Proteobacteria | Alphaproteobacteria | Rhodovibrionales | Kiloniellaceae | Kiloniellaceae_unclassified | otu519 | 3 | 0.1 | 0.0 |
| Proteobacteria | Alphaproteobacteria | Rhizobiales | Rhizobiaceae | Cohaesibacter | otu71 | 4 | 0.7 | 0.1 |
| Actinobacteria | Acidimicrobiia | Actinomarinales | Actinomarinales_unclassified | Actinomarinales_unclassified | otu163 | 2 | 0.3 | 0.1 |
| Actinobacteria | Acidimicrobiia | Actinomarinales | Actinomarinales_unclassified | Actinomarinales_unclassified | otu532 | 6 | 0.2 | 0.0 |
| Acidobacteria | Subgroup 9 | Subgroup 9_unclassified | Subgroup 9_unclassified | Subgroup 9_unclassified | otu342 | 3 | 0.3 | 0.0 |
| Acidobacteria | Subgroup 9 | Subgroup 9_unclassified | Subgroup 9_unclassified | Subgroup 9_unclassified | otu674 | 3 | 0.1 | 0.0 |
| Gemmatimonadetes | Gemmatimonadetes | Gemmatimonadales | Gemmatimonadaceae | Gemmatimonadaceae_unclassified | otu203 | 1 | 0.4 | 0.2 |
| Proteobacteria | Alphaproteobacteria | Kordiimonadales | Kordiimonadaceae | Kordiimonas | otu86 | 2 | 0.4 | 0.1 |
| Bacteroidetes | Bacteroidia | Cytophagales | Cyclobacteriaceae | Cyclobacteriaceae_unclassified | otu233 | 3 | 0.4 | 0.0 |
| Dadabacteria | Dadabacteriia | Dadabacteriales | Dadabacteriales_unclassified | Dadabacteriales_unclassified | otu347 | 3 | 0.2 | 0.0 |
| Dadabacteria | Dadabacteriia | Dadabacteriales | Dadabacteriales_unclassified | Dadabacteriales_unclassified | otu1016 | 3 | 0.1 | 0.0 |
| Actinobacteria | Thermoleophilia | Solirubrobacterales | 67-14 | 67-14_unclassified | otu324 | 3 | 0.3 | 0.0 |
| Proteobacteria | Deltaproteobacteria | NB1-j | NB1-j_unclassified | NB1-j_unclassified | otu344 | 1 | 0.1 | 0.0 |
| Planctomycetes | Planctomycetacia | Pirellulales | Pirellulaceae | Pirellulaceae_unclassified | otu538 | 2 | 0.1 | 0.0 |
| Actinobacteria | Acidimicrobiia | Microtrichales | Microtrichaceae | Microtrichaceae_unclassified | otu669 | 2 | 0.1 | 0.0 |
| Acidobacteria | Blastocatellia (Subgroup 4) | Blastocatellales | Blastocatellaceae | Blastocatella | otu489 | 3 | 0.1 | 0.0 |
| Entotheonellaeota | Entotheonellia | Entotheonellales | Entotheonellaceae | Entotheonellaceae_unclassified | otu788 | 4 | 0.1 | 0.0 |
| Acidobacteria | Thermoanaerobaculia | Thermoanaerobaculales | Thermoanaerobaculaceae | Subgroup 10 | otu711 | 3 | 0.1 | 0.0 |
| Proteobacteria | Gammaproteobacteria | Oceanospirillales | Kangiellaceae | Kangiellaceae_unclassified | otu744 | 5 | 0.1 | 0.0 |
| Proteobacteria | Gammaproteobacteria | Thiohalorhabdales | Thiohalorhabdaceae | Thiohalorhabdaceae_unclassified | otu571 | 6 | 0.1 | 0.0 |
| Bacteroidetes | Bacteroidia | Cytophagales | Cyclobacteriaceae | Ekhidna | otu651 | 5 | 0.1 | 0.0 |
| Bacteroidetes | Bacteroidia | Flavobacteriales | Flavobacteriaceae | Flavobacteriaceae_unclassified | otu579 | 6 | 0.1 | 0.0 |
| Gemmatimonadetes | BD2-11 terrestrial group | BD2-11 terrestrial group_unclassified | BD2-11 terrestrial group_unclassified | BD2-11 terrestrial group_unclassified | otu1439 | 3 | 0.1 | 0.0 |

(b)





| OTU | NCBI ID ≥ 99% similarity | Habitat(s) |
|---|---|---|
| otu29 | KT748605.1; JX227334.1; EU491654.1 | basaltic crust; nodule fields |
| otu11 | JX227511.1; JQ013353.1; FJ938664.1 | nodule fields; deep-sea sediments; cobalt-rich crust |
| otu31 | MG580220.1; KF268757.1 | Mariana subduction zone sediments; heavy metal contaminated marine sediments |
| otu83 | MG580220.1; JN621543.1 | Mariana subduction zone sediments; manganese oxide-rich marine sediments |
| otu160 | MG580740.1; JX227257.1 | Mariana subduction zone sediments; nodule fields |
| otu249 | JQ287236.1; KM051824.1 | inactive hydrothermal sulfides; basaltic crust |
| otu66 | JX226721.1 [a] | nodule fields |
| otu78 | JN860354.1; HQ721444.1 | hydrothermal vents; deep-sea sediments; |
| otu202 | MG580143.1; JX227690.1; JN860358.1 | Mariana subduction zone sediments; nodule fields; hydrothermal vents |
| otu317 | JX227432.1; AY627518.1 | nodule fields; deep-sea sediments; |
| otu947 | JX226721.1 [a] | nodule fields |
| otu588 | LC081043.1 | nodule |
| otu425 | JX227680.1; FJ938661.1 | nodule fields; cobalt-rich crust |
| otu68 | JN886931.1; FJ752931.1; KJ590663.1 | hydrothermal carbonate sediments; polychaete burrow environment; biofilm |
| otu227 | MG580382.1; AM997732.1 | Mariana subduction zone sediments; deep-sea sediments |
| otu215 | KC901562.1; AB015560.1 | basaltic glasses; deep-sea sediments |
| otu636 | HM101002.1; EU491612.1; KC682687.1 | Marine Sponge Halichondria; ocean crust; |
| otu434 | EU287401.1; JN977323.1 | Subsurface sediments; marine sediments |
| otu36 | JX227383.1; KY977840.1; AM997938.1 | nodule fields; Mariana subduction zone sediments; deep-sea sediments |
| otu162 | FN553503.1; AM997671.1 | hydrothermal vents; deep-sea sediments |
| otu97 | JX227693.1; FJ024322.1; EU491736.1 | nodule fields; ocean crust |
| otu266 | AB694157.1; JX227083.1 | deep-sea benthic foraminifera; nodule fields |
| otu521 | KY977757.1; KT336088.1; JX227223.1 | Mariana subduction zone sediments; nodules; nodule fields |
| otu346 | KY977757.1; JX227223.1 | Mariana subduction zone sediments; nodule fields |
| otu991 | JX227363.1; AM997733.1 | nodule fields; deep-sea sediments |
| otu113 | JX226757.1; EU491557.1 | nodule fields; ocean crust |
| otu184 | EU491404.1 | ocean crust |
| otu234 | EU491604.1 | ocean crust |
| otu255 | JX227709.1; FJ437705.1; KM110219.1 | nodule fields; hydrothermal deposits |
| otu122 | MG580277.1; AM997814.1; AJ966605.1 | Mariana subduction zone sediments; deep-sea sediments; nodule fields |
| otu326 | JN886905.1; KT748584.1 | hydrothermal carbonate sediments; basalt crust |
| otu865 | JX227375.1; FJ938651.1; AY225640.1 | nodule fields; cobalt-rich crust; hydrothermal sediments |
| otu171 | AM997407.1; FJ205352.1; EU491267.1 | deep-sea sediments; hydrothermal vents; ocean crust |
| otu541 | AB694393.1 | deep-sea benthic foraminifera |
| otu357 | EU236317.1; GU302472.1 | marine sponge; hydrocarbon seep |
| otu435 | KY609381.1; KM051717.1; JX226899.1 | Fe-rich hydrothermal deposits; basaltic crust; nodule fields |
| otu370 | EU491648.1 [a] | ocean crust |
| otu467 | FN553612.1; AB858542.1; KM051770.1 | hydrothermal vents; sulfide deposits; basaltic crust |
| otu450 | AM997745.1; KM051762.1; EU491108.1 | deep-sea sediments; basaltic crust; ocean crust |
| otu519 | GU220747.1; MG580729.1 | Fe-rich hydrothermal deposits; Mariana subduction zone sediments |
| otu71 | FJ205181.1; JX226787.1 | hydrothermal vents; noduel fields |
| otu163 | JX227427.1; JN886907.1; EU491661.1 | nodule fields; hydrothermal carbonate sediments; ocean crust |
| otu532 | EU491402.1; JX227188.1; EU374100.1 | ocean crust; nodule fields; deep-sea sediments |
| otu342 | JX227410.1; FJ205219.1; KT336055.1 | nodule fields; hydrothermal vents; nodules |
| otu674 | JX227662.1; KT336085.1; FJ938601.1 | nodule fields; nodules; cobalt-rich crust |
| otu203 | KP305065.1; FJ938598.1 | corals; cobalt-rich crust |
| otu86 | AM997620.1; FJ938474.1 | deep-sea sediments; cobalt-rich crust |
| otu233 | JX227464.1; AM997441.1 | nodule fields; deep-sea sediments |
| otu347 | JX227062.1; EU491655.1 | nodule fields; ocean crust |
| otu1016 | KF616695.1; KM396663.1; EU491261.1 | carbonate methane seep; brine seep; ocean crust |
| otu324 | JX226791.1; JN886912.1 | nodule fields; hydrothermal carbonate sediments |
| otu344 | EU438185.1; KY977824.1 | deep-sea sediments and hydrohtermal vents; ocean crust |
| otu538 | KM356353.1; JX226930.1; DQ996924.1 | carbonate methane seep; nodule fields; deep-sea sediments |
| otu669 | EU491619.1; MG580068.1; KT748607.1 | ocean crust |
| otu489 | EU491660.1; MG580531.1; AM998023.1 | ocean crust; deep-sea sediments |
| otu788 | JN886890.1; MG580099.1 | hydrothermal carbonate sediments; ocean crust |
| otu711 | JX193423.1; GU302449.1; AY225643.1 | mariculture sediments; hydrocarbon seep; ocean crust |
| otu744 | AB831375.1; EU290406.1; KM454306.1 | deep-sea methane-seep sediments; marine sponge; marine sediments |
| otu571 | JQ287033.1; AM911385.1; EU236424.1 | hydrothermal sulfides; cold-water corals; sponges |
| otu651 | KT972875.1 [a] | outcrops |
| otu579 | EU491573.1; KT336070.1 | ocean crust; nodules |
| otu1439 | JN886922.1; KC747092.1; JN884864.1 | hydrothermal carbonate sediments; deep-sea sediments; methane seep |


[a] ≥ 98% similarity.