# Peer review of "The contribution of microbial communities in"

_Biogeosciences, 2020_

## Referee Comment (RC1) · Beth Orcutt (Referee) · 25 Feb 2020

Review by Beth N. Orcutt and Tim D'Angelo, Bigelow Laboratory for Ocean Sciences

This study documents the composition and relative diversity of bacterial and archaeal microbial communities inhabiting polymetallic nodules and surrounding sediment of the Peru Basin collected in 2015. The motivations for this study are to determine if poly-metallic nodules have unique microbial communities, as such seabed mineral deposits may be targeted for deep-sea mining. While there have been similar prior studies of microbial community composition of polymetallic nodules, those studies focused on areas in the northern and central Pacific Ocean where organic carbon deposition rates are

lower. Thus, the new study from closer to an equatorial region with higher organic carbon export rates allows an analysis of how broader oceanographic properties impact microbial community diversity.

The first major claim of the current study is that microbial diversity is higher in the surrounding sediment than in the polymetallic nodules. This finding is different from a recent survey of available data from polymetallic nodules and sediments of the comparable Clairon Clipperton Zone, which indicated that nodules and sediments had comparable levels of diversity: https://ran-s3.s3.amazonaws.com/isa.org.jm/s3fs-public/files/documents/deep_ccz_biodiversity_synthesis_workshop_report_-_final.pdf. We encourage the authors to consider the implications of these differences between studies, and if data processing steps could be part of this difference.

Related to this part of this study, we caution that the workflow described in the methods may lead to inflated diversity metrics. The workflow described in L143-144 may allow lower quality sequence reads to pass the QC step, as most published workflows don't allow for sliding window PHRED scores of less than 28-30. For example, Dorado Outcrop basalt samples have around 1500 OTUs after filtering out low abundance/prevalence OTUs (described in Lee et al., 2016). We would expect a similar diversity on nodule samples exposed to bottom seawater but the samples described in this study have 5 - 14K OTUs per sample. Low quality reads can result in artificially large number of OTUs when using clustering-based methods. This has been documented by the developers of MOTHUR as a problem with low quality reads associated with old problematic Illumina chemistry kits. Even if there are true biological differences between Dorado Outcrop basalts and the samples in the current study that translate to different alpha diversity patterns, the presence of 525,169 singletons (as seen in Table 2) is a sign that there are likely issues with the QC steps of this workflow. We recommend that the authors revisit the sequence processing steps and consider using higher quality thresholds, and also consider using an algorithm that produces unique sequence variants (i.e. ASVs) instead of OTU clustering. Moreover, we wonder if there

is a more streamlined way to present the information included in Figure 1, or if some of this information could be moved to supplemental materials? It seems like a bit of overkill to have 10 plots essentially showing the same information.

A second major effort of this work is to identify taxa that are differentially abundant between nodules and sediments. While the text in Lines 244-261 describes these differences, and Table 4 includes the result of Aldex2 analysis, we don't find that Figure 4 visually conveys these differences in an easily digestible way and suggest using differential log abundance plots to more clearly show which taxa vary between the sample types.

Another major focus of this work is the comparison of the microbial community structures between the Peru Basin nodules and those of the CCZ. I think that the paper could be improved by providing some kind of summary graphic or schematic that visually explains the differences, and their causes, as described in the text. For example, a cartoon illustrating that the lower OC flux in the CCZ leads to nodules that look like X with communities that look like Y and perform Z functions, versus how those conditions are different at the Peru Basin. Such a summary graphic could really help simplify the presentation of the major recommendations from this work in a way that is easy to grasp, which will be especially helpful for policy makers thinking about deep-sea mining.

A question: in the methods, there is mention of collecting samples for cell abundance determination, but such data are not presented in this paper. Is it possible to include such data? This would help to evaluate if the "hot spot" idea discussed in the paper correlates to cell biomass - i.e. is lower diversity correlated to higher biomass?

A suggestion: there is some mismatch between the 3 hypotheses posed in the introduction and the three objectives posed in the discussion section. The discussion text follows the outline of the objectives, but there is not explicit "testing" of the hypotheses proposed at the beginning of the paper, and also the discussion does exactly follow the

objectives as proposed. For example, discussion section 4.3 discusses metabolisms inferred from the amplicon data, not what environmental factors structure the community, as would be assumed by how objective three is worded. We recommend bringing better alignment between the hypotheses/objectives and what the data actually address.

minor suggestions:

L1: Title could be more descriptive of what the study discovered

L15 - consider removing "need to"

L22 - Acidomicrobia, only one "i". To update throughout the manuscript.

L78-79 - need consistency in the presentation of thousands of kilometers. In one instance, there is no punctuation; in the second instance, there is punctuation.

L80 - missing a decimal point in 0.2-0.6%?

L123 - were any negative DNA extraction controls included in this study, since low biomass might have been expected? If yes, please describe.

L141 - Is there a reference that shows why these trimmomatic SLIDINGWINDOW parameters were used? They seem relaxed and would allow for sub-par quality reads to pass the QC step. Most workflows don't allow for sliding window PHRED scores of less than 28-30.

L141 - recommendation to deposit your data processing pipeline to github or similar repository.

L144 - There is a comparative "while" statement describing the differences between how bacterial and archaeal sequences were merged, but the way it is worded, it appears to describe the same order of operations.

L174 - Transforming count matrices using the center-log ratio requires a strategy for replacing zeros with a pseudo count because the presence of zeros produces NA values. There is no zero-replacement strategy described in this workflow. The Bray-Curtis distance cannot be computed on data matrices that contain negative numbers. A Center-log-Ratio transformed count matrix contains negative numbers. CLR transformed data is usually ordinated using the Aitchison distance metric or the Euclidean distance. I am unclear on how these analyses were performed in the way that they are described. Was the data log10(x+1) transformed? That transformation is compatible with the bray-curtis distance. The resulting ordinations looks correct, but I think the description in the methods section is inaccurate. Could the authors provide a document with the code used to perform these steps?

L237 - these percentages are for all nodules in aggregate as an average, but does not show the variation between samples. I recommend including standard deviation plus/minus for each percentage.

L338 - could the differences in relative percentages of archaea between this study and prior studies be due to difference in DNA extraction, primers used, or sequencing approach?

L359 - suggestion to add a clause to the end of the sentence regarding nodules and sediments have distinct communities, stating that this observation is consistent with what has been found in earlier studies, and cite a few examples.

L413 - "reductive"

---

## Referee Comment (RC2) · Anonymous Referee #2 · 6 Mar 2020

General comments to authors:

The manuscript by Molari et al. describes the microbial community structure associated with sediments and manganese nodules from 3 and 2 sites, respectively, within the Peru Basin.

The authors find that Gammaproteobacteria and Alphaproteobacteria are the dominant bacterial classes in sediments and manganese nodules while all archaeal communities investigated were dominated by Thaumarchaeota. However, sediment and nodule communities were found to differ significantly at the OTU level, as assessed by calculating Jaccard dissimilarity. The authors note differences in the nodule community

composition (specifically, a lower relative abundance of Archaea, and a different nitrifier community) in their study in the Peru Basin as compared with communities in the Clarion-Clipperton Fracture Zone (CCZ), where previous work on microbial community composition of nodules has been done.

The strengths of the manuscript include the following:

i. There is a lack of studies of the prokaryotic diversity in the surface sediments and nodules of the Peru Basin, which has different environmental conditions than the relatively well-studied CCZ. ii. The molecular and bioinformatic methods are well-documented and the microbial community analysis is thorough.

Weaknesses of the manuscript include the following:

i. The lack of metadata associated with the various sites makes interpretation of the differing community structures among sites difficult.

Specific comments to the authors:

Major concerns:

1. Page 3 – 4. Somewhere in this discussion of the CCZ versus the Peru Basin I think it would be helpful to briefly let the reader know the state of hypothetical mining in each of these regions. In the CCZ, the ISA has entered into contracts with various contractors for exploration for polymetallic nodules. Is this the case in the Peru Basin as well?

2. Page 5, line 113. "Samples were collected at three sites. . ." For clarity I think the authors should explicitly state in the text that nodules were only collected at 2 of these 3 sites.

3. Page 5, line 115. ". . . called "Reference Sites." I suggest directly listing the Reference Sites here in the text instead of making the reader consult Table 1, especially since the authors refer to Reference South later in the text. Could change to ". . . called

"Reference Sites": Reference East, Reference West, and Reference South."

4. Page 5, line 116. Here a map of the Peru Basin (in addition to the Table already provided), with the study sites and DISCOL experiment sites marked, would be very helpful to the reader.

5. Page 8, lines 226 – 232. "...significant differences were detected in sediment microbial community structure among the different sites... "Site" defined by geographic location and "Substrate" ... explained a similar proportion of variation in bacterial community structure..." This was a bit surprising to me and this is where I think some physical/chemical/biological metadata about each site would be really helpful. If any is available, perhaps from other groups on the cruise, it would help add context to some of the observations here.

6. Page 8, lines 226 – 229. "...significant differences were detected in sediment microbial community structure ... between communities associated with nodules and sediments at Reference South." I think it is important to state directly in the text that this site, Reference South, was the only site that had enough nodule sampling to allow the authors to do this analysis (at least I assume this is what occurred). Otherwise this sentence could be taken to mean that differences in community structure between nodules and sediments were also investigated at the other 2 sites, and no differences were found.

Minor issues to be addressed:

7. Page 8, line 238. "Aphaproteobacteria" should be "Alphaproteobacteria".

8. Page 9, line 282. "Aphaproteobacteria" should be "Alphaproteobacteria".

―――――――――――――――――――――

---

## Author Comment (AC1) · 4 Apr 2020

Authors' Response ('AR') to interactive comment on "Microbial communities associated with sediments and polymetallic nodules of the Peru Basin" by Massimiliano Molari et al.

Review by Beth N. Orcutt and Tim D'Angelo ('RC1')

RC1> This study documents the composition and relative diversity of bacterial and archaeal microbial communities inhabiting polymetallic nodules and surrounding sediment of the Peru Basin collected in 2015. The motivations for this study are

to determine if polymetallic nodules have unique microbial communities, as such seabed mineral deposits may be targeted for deep-sea mining. While there have been similar prior studies of microbial community composition of polymetallic nodules, those studies focused on areas in the northern and central Pacific Ocean where organic carbon deposition rates are lower. Thus, the new study from closer to an equatorial region with higher organic carbon export rates allows an analysis of how broader oceanographic properties impact microbial community diversity. The first major claim of the current study is that microbial diversity is higher in the surrounding sediment than in the polymetallic nodules. This finding is different from a recent survey of available data from polymetallic nodules and sediments of the comparable Clairon Clipperton Zone, which indicated that nodules and sediments had comparable levels of diversity: https://ran-s3.s3.amazonaws.com/isa.org.jm/s3fs-public/files/documents/deep_ccz_biodiversity_synthesis_workshop_report_-_final.pdf. We encourage the authors to consider the implications of these differences between studies, and if data processing steps could be part of this difference.

AR> We thank the Reviewers for their thorough and very helpful revision, and for pointing us to the results of the recent meta-analysis of microbial diversity data available for CCZ, which was not available at the moment of submission.

In the revised MS we will include the outcome of the workshop in the discussion by adding the following statement (added/replaced text in italics): 'Microbial communities associated to nodules are significantly less diverse than those in the sediments, and the decrease in diversity was observed both in rare and abundant bacterial types (Figure 1 and S1). This seems to be a common feature of polymetallic nodules (Wu et al., 2013; Tully and Heidelberg, 2013; Zhang et al., 2014; Shulse et al., 2016; Lindh et al. 2017). However, a recent meta-analysis of 16S rRNA gene diversity reports no significant differences between microbial biodiversity between nodules and sediments within the studied habitats (Church et al., 2019). Church and colleagues also pointed out that the findings are so far not conclusive due to the limited number of studies and

differences in methods (e.g. PCR primers, sequencing approaches) which may also be a reason for the differences between the meta-analysis and the results of this study.'

RC1> Related to this part of this study, we caution that the workflow described in the methods may lead to inflated diversity metrics. The workflow described in L143-144 may allow lower quality sequence reads to pass the QC step, as most published workflows don't allow for sliding window PHRED scores of less than 28-30. For example, Dorado Outcrop basalt samples have around 1500 OTUs after filtering out low abundance/prevalence OTUs (described in Lee et al., 2016). We would expect a similar diversity on nodule samples exposed to bottom seawater but the samples described in this study have 5 - 14K OTUs per sample. Low quality reads can result in artificially large number of OTUs when using clustering-based methods. This has been documented by the developers of MOTHUR as a problem with low quality reads associated with old problematic Illumina chemistry kits. Even if there are true biological differences between Dorado Outcrop basalts and the samples in the current study that translate to different alpha diversity patterns, the presence of 525,169 singletons (as seen in Table 2) is a sign that there are likely issues with the QC steps of this workflow. We recommend that the authors revisit the sequence processing steps and consider using higher quality thresholds, and also consider using an algorithm that produces unique sequence variants (i.e. ASVs) instead of OTU clustering. Moreover, we wonder if there is a more streamlined way to present the information included in Figure 1, or if some of this information could be moved to supplemental materials? It seems like a bit of overkill to have 10 plots essentially showing the same information.

AR> We thank the Reviewers for the opportunity to clarify the bioinformatics workflow. We recognize that how it is reported in "Methods" of the MS may be misleading. As a standard procedure, we applied a score of 10 for bacteria and 13 for Archaea in quality trimming, but then the quality of sequences was assessed with the software FastQC (http://www.bioinformatics.babraham.ac.uk/projects/fastqc/). If the sequences did not pass the quality check, then they were filtered again with an appropriate quality score.

[Figure]

All sequences used in the MS successfully passed the FastQC quality control, with average quality score per sample >34 for Bacteria, and >22 for Archaea. Thus, we believe that the high numbers of OTUs per sample was not caused by the introduction of low-quality sequences in the analysis.

In the revised MS the sequences workflow will be clarified as follows (added/replaced text in italics): 'Subsequently the TRIMMOMATIC software (Bolger et al., 2014) was used to remove low-quality sequences starting with the following settings: SLIDING-WINDOW:4:10 MINLEN:300 (for Bacteria); SLIDINGWINDOW:6:13 MINLEN:450 (for Archaea). In case of bacteria data this step was performed before the merging of reverse and forward reads with PEAR (Zhang et al., 2014). Merging of the archaeal reads was done before removing low-quality sequences in order to enhance the number of retained reads due to long archaeal 16S fragments. All sequences were quality controlled with FastQC (Andrews, 2010). Where necessary, more sequences were removed with TRIMMOMATIC with larger sliding window scores until the FastQC quality control was passed (average quality score per sample >34 for Bacteria and >22 for Archaea).'

We agree that the ASV approach has a higher taxonomic resolution than OTU clustering. However, for the purpose of this paper the resolution returned by SWARM (i.e. "species" level) appears appropriate, as it allowed to distinguish microbial communities associated with nodules and sediments (i.e. Figure 2). Furthermore, according to our experience in other studies >90% OTUs generated by SWARM overlap with variants (ASVs) identified with Dada2.

Regarding Figure 1, we do not fully agree with the Reviewers' view that the panels repeatedly show the "same information". Diversity indices and unique OTUs are reported for Bacteria and Archaea in upper and lower panels, respectively – hence the upper and lower rows of panels refer to independent data sets. As the reviewers are certainly aware, the diversity indices presented in the first four plots or each row differ in their ecological meaning: i) total number of OTUs (H0) provide overall information

about alpha-diversity, ii) exponential Shannon (H1) considers species richness and equitability, iii) inverse Simpson (H2) accounts for dominant taxa, and iv) chao1 accounts for rare taxa. Here this was calculated with the same number of sequences for each sample, thus it is not affected by sequencing depth. The last plot shows the contribution of unique OTUs to the total number of OTUs and, hence again has a different focus. While the pattern shown in the different panels may be visually similar, we are still convinced that each panel contains important information and should be presented to the reader.

Therefore, we would like keep the figure in the revised version of the MS. Upon specific request by the editor we would, however move plots for Archaea to the supplementary information. They contribute only minor to the total diversity as compared to Bacteria, but we would stick to the full set of panels for Bacteria.

RC1> A second major effort of this work is to identify taxa that are differentially abundant between nodules and sediments. While the text in Lines 244-261 describes these differences, and Table 4 includes the result of Aldex2 analysis, we don't find that Figure 4 visually conveys these differences in an easily digestible way and suggest using differential log abundance plots to more clearly show which taxa vary between the sample types.

AR> We thank the reviewers for sharing their thoughts about improving the data representation in Fig. 4. In the revised MS Figure 4 will be replaced by a fold- change plot showing genera enriched in nodules compared to those found in sediments.

RC1> Another major focus of this work is the comparison of the microbial community structures between the Peru Basin nodules and those of the CCZ. I think that the paper could be improved by providing some kind of summary graphic or schematic that visually explains the differences, and their causes, as described in the text. For example, a cartoon illustrating that the lower OC flux in the CCZ leads to nodules that look like X with communities that look like Y and perform Z functions, versus how

those conditions are different at the Peru Basin. Such a summary graphic could really help simplify the presentation of the major recommendations from this work in a way that is easy to grasp, which will be especially helpful for policy makers thinking about deep-sea mining.

AR> We appreciate the suggestion of the reviewers and we generally agree with them. However, such a scheme would need to be supported by a deeper analysis that would require a more comprehensive dataset. Such a generalized analysis unfortunately cannot be carried out at the moment due to limited number of studies and different sequencing methods applied to investigate microbial communities between nodules fields. The aims of this study were to explore the role of nodules in deep-sea microbial diversity, their potential role in ecosystem functions, and a comparison of our results with data available from other nodule field regions (i.e. CCZ). Our results highlight the importance of nodules in hosting specific and potentially functional important microbes, which differ from those reported for CCZ. While the number of samples and differences in methods do not allow a generalization, we felt the need to point out important ecological questions and hypotheses that are relevant in the deep-sea mining context, but that are not yet solved and should be addressed in future studies.

This consideration will be highlighted in "Conclusions" of the revised MS by adding the following statement (added/replaced text in italics): 'However remarkable differences in community composition (e.g. Mn-cycling bacteria, nitrifiers) between the CCZ and the Peru Basin in microbial community composition also show that environmental settings (i.e. POC flux) and features of FeMn nodules (e.g. metal content) may play a significant role in structuring the nodule microbiome. Due to limitations in the available datasets and methodological differences in the studies existing to date, findings are not yet conclusive and cannot be generalized. However, they indicate that microbial community structure and function would be impacted by nodule removal. Future studies need to look at these impacts in more detail and need to address regional differences, to determine the spatial turnover and its environmental drivers, and the consequences

regarding endemic types.'

RC1> A question: in the methods, there is mention of collecting samples for cell abundance determination, but such data are not presented in this paper. Is it possible to include such data? This would help to evaluate if the "hot spot" idea discussed in the paper correlates to cell biomass - i.e. is lower diversity correlated to higher biomass?

AR> Unfortunately, we do not have cell counts for manganese nodules. Originally, we planned to include cell counts (AODC and CARD-FISH) for sediments. However, these data are reported already in another study on the impact of mining on sediment microbial communities and their biogeochemical functions which was under revision at the moment of MS submission but will be available soon (Vonnahme T.R, Molari M., Janssen F., Wenzhöfer F., Haeckel M., Titschack J., Boetius A. Effects of a deep-sea mining experiment on seafloor microbial communities and functions after 26 years. Science Advances, in press). We will clarify this issue in the revised MS and point the reader to the Vonnahme et al. publication.

RC1> A suggestion: there is some mismatch between the 3 hypotheses posed in the introduction and the three objectives posed in the discussion section. The discussion text follows the outline of the objectives, but there is not explicit "testing" of the hypotheses proposed at the beginning of the paper, and also the discussion does exactly follow the objectives as proposed. For example, discussion section 4.3 discusses metabolisms inferred from the amplicon data, not what environmental factors structure the community, as would be assumed by how objective three is worded. We recommend bringing better alignment between the hypotheses/objectives and what the data actually address.

AR> We thank the reviewers for pointing out these inconsistencies.

The first and second hypothesis (Hp1 and Hp2) have been tested using statistical tests (as described in the Methods section). Hypotheses and "primary aims/objectives" are discussed in section 4.1 and 4.2, respectively. The "secondary aim" (previ-
ously Hp3) was addressed by deducing potential metabolic functions and habitat features/preferences from the taxa that were significantly enriched in the nodules (based on statistical testing) and what we know from descriptions of closely related organisms. From these results and comparison with CCZ microbial data we suggested potential environmental factors that can have a major role in shaping the microbial community on nodules. Based on the limited available data, however, these suggestions cannot be rigorously tested. The "secondary objective" was mainly discussed in section 4.3, but also partially addressed in the previous two sections.

In the revised MS we will improve the alignment of hypotheses/aims with objectives by slightly refocusing the first and second hypothesis (Hp1 and Hp2) and by turning the third hypothesis into a secondary aim.

1) Hp1 [introduction]: 'nodules shape deep-sea microbial diversity and functions'

Primary Objective [discussion part 4.1]: compares the microbes of nodule fields with microbiota of deep-sea sediments in other ecosystems in order to identify specific features of microbial diversity of nodule fields.

2) Hp2 [introduction]: 'nodules host a specific microbial community and functions compared to the surrounding sediments'

Primary Objective [discussion part 4.2]: elucidates differences in diversity and in microbial community structure between sediments and nodules, and the potential implications for microbially-mediated functions

3) Secondary aim [introduction]: 'Another aim of this study was to investigate the nodule features that may play a major role in shaping microbial community composition.'

Secondary objective [discussion primarily part 4.3]: investigates the major drivers in shaping microbial communities associated with nodules

The modified hypotheses / aims are easily associated with the titles of sections 4.1, 4.2, and 4.3 of the discussion but we will make sure to refer back to the hypotheses /

aims also in the text.

RC1> minor suggestions:

RC1> L1: Title could be more descriptive of what the study discovered

AR> According to reviewers' suggestion, the title of revised MS will be: "The contribution of microbial communities in polymetallic nodules to the diversity of the deep-sea microbiome of Peru Basin (4130 − 4198 meter depth)"

RC1> L15 - consider removing "need to"

AR> We will remove "need to".

RC1> L22 - Acidomicrobia, only one "i". To update throughout the manuscript.

AR> We are referring here to the class Acidimicrobiia within the phylum Actinobacteria in accordance with the NCBI taxonomy (https://www.ncbi.nlm.nih.gov/Taxonomy/Browser/wwwtax.cgi?id=84992)

RC1> L78-79 - need consistency in the presentation of thousands of kilometers. In one instance, there is no punctuation; in the second instance, there is punctuation.

AR> We will correct this by removing punctuation.

RC1> L80 - missing a decimal point in 0.2-0.6%?

AR> Reviewers are correct - we will change this accordingly.

RC1> L123 - were any negative DNA extraction controls included in this study, since low biomass might have been expected? If yes, please describe.

AR> Yes, we had negative controls. This is mentioned in lines 129-130 and table 2 of the submitted version of the MS.

RC1> L141 - Is there a reference that shows why these trimmomatic SLIDINGWINDOW parameters were used? They seem relaxed and would allow for sub-par quality

reads to pass the QC step. Most workflows don't allow for sliding window PHRED scores of less than 28-30.

AR> We give detailed information above in the general comments section, and we will provide additional information the revised MS.

RC1> L141 - recommendation to deposit your data processing pipeline to github or similar repository.

AR> We are currently exploring this with experts in our group. Once the code is made available we will add the information to the revised manuscript.

RC1> L144 - There is a comparative "while" statement describing the differences between how bacterial and archaeal sequences were merged, but the way it is worded, it appears to describe the same order of operations.

AR> We agree and will correct in the revised version of the manuscript as follows (added/replaced text in italics): 'In case of bacteria data this step was performed before the merging of reverse and forward reads with PEAR (Zhang et al., 2014). Merging of the archaeal reads was done before removing low-quality sequences in order to enhance the number of retained reads due to long archaeal 16S fragments.'

RC1> L174 - Transforming count matrices using the center-log ratio requires a strategy for replacing zeros with a pseudo count because the presence of zeros produces NA values. There is no zero-replacement strategy described in this workflow. The Bray-Curtis distance cannot be computed on data matrices that contain negative numbers. A Center- log-Ratio transformed count matrix contains negative numbers. CLR transformed data is usually ordinated using the Aitchison distance metric or the Euclidean distance. I am unclear on how these analyses were performed in the way that they are described. Was the data log10(x+1) transformed? That transformation is compatible with the bray- curtis distance. The resulting ordinations looks correct, but I think the description in the methods section is inaccurate. Could the authors provide a document

with the code used to perform these steps?

AR> Totally right, indeed the Euclidean distance matrix was used and not Bray-Curtis (as also specified in caption of Figure 2). In the revised MS we will correct this mistake as follows (added/replaced text in italics): Beta-diversity in samples from different substrates and from the substrate in samples from different sites was quantified by calculating an Euclidean distance matrix based on centred log-ratio (CLR) transformed OTU abundances (function clr in R package compositions) and Jaccard dissimilarity based on a presence/absence OTU table.

RC1> L237 - these percentages are for all nodules in aggregate as an average, but does not show the variation between samples. I recommend including standard deviation plus/minus for each percentage.

AR> The percentage reported is not the average between/within groups, but it is the result of hierarchical clustering (function hclust in R package vegan) using the complete linkage method (data reported in Figure 4) and, hence, standard deviation cannot be reported. This information will be added to the revised MS as follows (added/replaced text in italics): The Jaccard dissimilarity coefficient was used to perform hierarchical clustering (function hclust in R package vegan, using the complete linkage method), and the dissimilarity values for cluster nodes were used to calculate the number of shared OTUs between/within groups.

RC1> L338 - could the differences in relative percentages of archaea between this study and prior studies be due to difference in DNA extraction, primers used, or sequencing approach?

AR> Previous data for Archaea are only reported by Tully and Heidelberg (2013) and Shulse at al. (2016) in the CCZ. In the first study a modified phenol-chloroform extraction method was applied for DNA extraction, universal primers (U515/U1048; targeting the V4 region of the 16S rRNA gene) for PCR amplification and Roche 454 Titanium platform for sequencing. Shulse and colleagues extracted DNA with the FastDNA Spin

Kit for Soil (MP Biomedicals, USA), PCR amplification was carried out with universal primers (515f/805r; targeting the V4 region of the 16S rRNA gene) and sequencing with illumina MiSeq platform. These pipelines indeed differ from those applied in our study and reported in the Methods section: DNA extraction with FastDNA Spin Kit for Soil (MP Biomedicals, USA), PCR with bacteria (341F/785R; targeting the V3-V4 region of the 16S rRNA gene) and archaea primers (349F/915R; targeting the V3-V5 region of the 16S rRNA gene), and sequencing with illumina MiSeq platform. We agree with the reviewers that different methods applied make the comparison difficult, especially with data from Tully and Heidelberg (2013) where differences in methodology appear most pronounced. In the revised MS we will limit the comparison to data from Shulse at al. (2016) because differences in methods are limited to the choice of primers, which however amplified the same hypervariable region of 16S rRNA gene (V4) reducing biases in the comparison. This, however, does not change the overall difference in relative percentages of archaea in this study compared to previous work. We will further add the statement that we cannot rule out that the slight differences in methodology between the Shulse et al. and our study could be a possible explanation for the observed differences.

RC1> L359 - suggestion to add a clause to the end− of the sentence regarding nodules and sediments have distinct communities, stating that this observation is consistent with what has been found in earlier studies, and cite a few examples.

AR> We will revise the MS accordingly (added/replaced text in italics): 'Analysis of community composition at OTU level shows that nodules and sediments host distinct bacterial and archaeal communities (Figure 2), as previously reported also for CCZ (Wu et al., 2013; Tully and Heidelberg, 2013; Shulse et al., 2016; Lindh et al. 2017).'

RC1> L413 - "reductive"

AR> We will replace "reducers" with "reductive"

---

## Author Comment (AC2) · 4 Apr 2020

Authors' Response ('AR') to the interactive comment on "Microbial communities associated with sediments and polymetallic nodules of the Peru Basin" by Massimiliano Molari et al.

Review by Anonymous Referee ('RC2')

RC2> General comments to authors: The manuscript by Molari et al. describes the microbial community structure associated with sediments and manganese nodules from 3 and 2 sites, respectively, within the Peru Basin.

The authors find that Gammaproteobacteria and Alphaproteobacteria are the dominant bacterial classes in sediments and manganese nodules while all archaeal communities investigated were dominated by Thaumarchaeota. However, sediment and nodule communities were found to differ significantly at the OTU level, as assessed by calculating Jaccard dissimilarity. The authors note differences in the nodule community composition (specifically, a lower relative abundance of Archaea, and a different nitrifier community) in their study in the Peru Basin as compared with communities in the Clarion-Clipperton Fracture Zone (CCZ), where previous work on microbial community composition of nodules has been done.

The strengths of the manuscript include the following: i. There is a lack of studies of the prokaryotic diversity in the surface sediments and nodules of the Peru Basin, which has different environmental conditions than the relatively well-studied CCZ. ii. The molecular and bioinformatic methods are well-documented and the microbial community analysis is thorough.

Weaknesses of the manuscript include the following: i. The lack of metadata associated with the various sites makes interpretation of the differing community structures among sites difficult.

AR> We appreciate the suggestion of the reviewer. However, the primary aim of this study was not to investigate and explain the variability of microbial community between sites, but between habitats (nodules and sediments). Only sedimentary metadata (e.g. pigments and organic carbon content, porewater profiles, and porosity) are available for sites investigated, and not for nodules, which precludes the quantitative characterization of nodule habitat setting. Thus sedimentary setting alone does not help to understand differences in microbial community structure and diversity that are observed between sediments and nodules. However, this metadata, as well as the discussion of variability of sedimentary environmental settings and microbial communities, will be soon (scheduled publication at the 29th of April) available in "Vonnahme T.R, Molari M., Janssen F., Wenzhöfer F., Haeckel M., Titschack J., Boetius A. Effects of a deepsea mining experiment on seafloor microbial communities and functions after 26 years. Science Advances, in press".

We point the reader to this publication in the revised version of the MS.

RC2> Specific comments to the authors:

RC2> Major concerns:

RC2> 1. Page 3 – 4. Somewhere in this discussion of the CCZ versus the Peru Basin I think it would be helpful to briefly let the reader know the state of hypothetical mining in each of these regions. In the CCZ, the ISA has entered into contracts with various contractors for exploration for polymetallic nodules. Is this the case in the Peru Basin as well?

AR> We will clarify this point in the revised MS modifying the introduction as follows (added/replaced text in italics): "Nodule accumulations of economic interest have been found in four geographical locations: the Clarion-Clipperton Fracture Zone (CCZ) and the Penrhyn Basin in the central north and south Pacific Ocean, respectively; the Peru Basin in the south-east Pacific; and in the center of the north Indian Ocean (Miller et al., 2018). *However to our knowledge there are no exploration activities and no plans for mining in the Peru basin so far.*"

RC2> 2. Page 5, line 113. "Samples were collected at three sites..." For clarity I think the authors should explicitly state in the text that nodules were only collected at 2 of these 3 sites.

AR> The Reviewer is right and we will clarify this issue in the revised MS as follows (added/replaced text in italics): "Manganese nodules where sampled, using a TV-MUC, or a Remotely Operated Vehicle (ROV Kiel6000): *one nodule at Reference West and four nodules at Reference South.*"

RC2> 3. Page 5, line 115. "... called "Reference Sites." I suggest directly listing the Reference Sites here in the text instead of making the reader consult Table 1, especially

since the authors refer to Reference South later in the text. Could change to "... called "Reference Sites": Reference East, Reference West, and Reference South."

AR> We will modify the revised MS according to the Reviewer's suggestion as follows (added/replaced text in italics): "Samples were collected at three sites outside the seafloor area selected in 1989 for a long-term disturbance and recolonization experiment (DISCOL; Thiel et al., 2001), *for this reason they were called "References Sites": Reference East, Reference West, and Reference South."*

RC2> 4. Page 5, line 116. Here a map of the Peru Basin (in addition to the Table already provided), with the study sites and DISCOL experiment sites marked, would be very helpful to the reader.

AR> An appropriate map is available in the Vonnahme et. al study mentioned above. We would suggest pointing the reader to that publication to avoid duplication. On request of the editor we are happy to provide a similar map for inclusion in the revised version of the MS.

RC2> 5. Page 8, lines 226 – 232. "... significant differences were detected in sediment microbial community structure among the different sites... "Site" defined by geographic location and "Substrate" ... explained a similar proportion of variation in bacterial community structure..." This was a bit surprising to me and this is where I think some physical/chemical/biological metadata about each site would be really helpful. If any is available, perhaps from other groups on the cruise, it would help add context to some of the observations here.

AR> A detailed environmental characterization of sites investigated and focused discussion of baseline condition (i.e. variability of environmental settings, and community structure and diversity between "Reference sites") will be soon available in the Vonnahme et al." study mentioned above. Primary aims of this study were: i) to compare the microbes of nodules fields with the microbiome of other deep-sea sediments, in order to identify specific features of microbial communities of nodule fields; ii) to elucidate

differences in diversity and in microbial community structure between sediments and nodules, and their potential implications for microbially-mediated functions. Thus, we believe that in order to achieve these aims it is neither needed nor beneficial to provide and discuss sedimentary metadata. However to meet demands of readers to better understand the effect of environmental settings on differences in microbial community structure between the sites investigated we will point out that these information can be found in "Vonnahme et al." in the revised MS.

RC2> 6. Page 8, lines 226 – 229. "...significant differences were detected in sediment microbial community structure ... between communities associated with nodules and sediments at Reference South." I think it is important to state directly in the text that this site, Reference South, was the only site that had enough nodule sampling to allow the authors to do this analysis (at least I assume this is what occurred). Otherwise this sentence could be taken to mean that differences in community structure between nodules and sediments were also investigated at the other 2 sites, and no differences were found.

AR> We thank the Reviewer for highlighting this point. We will clarify this issue in the revised MS as follows (added/replaced text in italics): 'Also, significant differences were detected in sediment microbial community structure among the different sites (PERMANOVA; Bacteria: R2 = 0.384; p = 0.003; F1,8 = 1.87; Archaea: R2 = 0.480; p = 0.013; F2,8 = 2.31; Table S1), and between communities associated with nodules and sediment at Reference South (PERMANOVA; Bacteria: R2 = 0.341; p = 0.023; F1,6 = 2.59; Archaea: R2 = 0.601; p = 0.029; F1,6 = 7.53; Table S1), which was the only site where the number of samples allowed for the test.'

RC2> Minor issues to be addressed:

RC2> 7. Page 8, line 238. "Aphaproteobacteria" should be "Alphaproteobacteria".

AR> It will be corrected.

RC2> 8. Page 9, line 282. "Aphaproteobacteria" should be "Alphaproteobacteria".

AR> It will be corrected.
* * *

---

## Author Response (AR1)

**Review by Beth N. Orcutt and Tim D'Angelo ('RC1')**

RC1> This study documents the composition and relative diversity of bacterial and archaeal microbial communities inhabiting polymetallic nodules and surrounding sediment of the Peru Basin collected in 2015. The motivations for this study are to determine if polymetallic nodules have unique microbial communities, as such seabed mineral deposits may be targeted for deep-sea mining. While there have been similar prior studies of microbial community composition of polymetallic nodules, those studies focused on areas in the northern and central Pacific Ocean where organic carbon deposition rates are lower. Thus, the new study from closer to an equatorial region with higher organic carbon export rates allows an analysis of how broader oceanographic properties impact microbial community diversity.

The first major claim of the current study is that microbial diversity is higher in the surrounding sediment than in the polymetallic nodules. This finding is different from a recent survey of available data from polymetallic nodules and sediments of the comparable Clairon Clipperton Zone, which indicated that nodules and sediments had comparable levels of diversity: https://ran-s3.s3.amazonaws.com/isa.org.jm/s3fs-public/files/documents/deep_ccz_biodiversity_synthesis_workshop_report_-_final.pdf. We encourage the authors to consider the implications of these differences between studies, and if data processing steps could be part of this difference.

AR> We thank the Reviewers for their thorough and very helpful revision, and for pointing us to the results of the recent meta-analysis of microbial diversity data available for the CCZ, which were not available at the moment of submission.

In the revised MS we included the outcome of the workshop in the discussion by adding the following statement (added/replaced text in *italics*):

'Microbial communities associated to nodules are *significantly* less diverse than those in the sediments, and the decrease in diversity was observed both in rare and abundant bacterial types (Figure 1 and S1). *This seems to be a common feature of polymetallic nodules (Wu et al., 2013; Tully and Heidelberg, 2013; Zhang et al., 2014; Shulse et al., 2016; Lindh et al. 2017). However, a recent meta-analysis of 16S rRNA gene diversity reports no significant differences in microbial biodiversity between nodules and sediments within the studied habitats in the CCZ (Church et al., 2019). Church and colleagues also pointed out that the findings are so far not conclusive due to the limited number of studies and differences in methods (e.g. PCR primers, sequencing approaches) which may also be a reason for the differences between the meta-analysis and the results of this study*.' [Lines 398-406]

RC1> Related to this part of this study, we caution that the workflow described in the methods may lead to inflated diversity metrics. The workflow described in L143-144 may allow lower quality sequence reads to pass the QC step, as most published workflows don't allow for sliding window PHRED scores of less than 28-30. For example, Dorado Outcrop basalt samples have around 1500 OTUs after filtering out low abundance/prevalence OTUs (described in Lee et al., 2016). We would expect a similar diversity on nodule samples exposed to bottom seawater but the samples described in this study have 5 - 14K OTUs per sample. Low quality reads can result in artificially large number of OTUs when using clustering-based methods. This has been documented by the developers of MOTHUR as a problem with low quality reads associated with old problematic Illumina chemistry kits. Even if there are true biological differences between Dorado Outcrop basalts and the samples in the current study that translate to different alpha diversity patterns, the presence of 525,169 singletons (as seen in Table 2) is a sign that there are likely issues with the QC steps of this workflow. We recommend that the authors revisit the sequence processing steps and consider using higher quality thresholds, and also consider using an algorithm that produces unique sequence variants (i.e. ASVs) instead of OTU clustering. Moreover, we wonder if there is a more streamlined way to present the information included in Figure 1, or if some of this information could be moved to supplemental materials? It seems like a bit of overkill to have 10 plots essentially showing the same information.

AR> We thank the Reviewers for the opportunity to clarify the bioinformatics workflow. We recognize that how it was reported in "Methods" of the original MS may have been misleading. As a standard procedure, we applied a score of 10 for bacteria and 13 for Archaea in quality trimming, but then the quality of sequences was assessed with the software FastQC (http://www.bioinformatics.babraham.ac.uk/projects/fastqc/). If the sequences did not pass the quality check, then they were filtered again with an appropriate quality score. All sequences used in the MS successfully passed the FastQC quality control, with an average quality score per sample >34 for Bacteria, and >22 for Archaea. Thus, we believe that the high numbers of OTUs per sample was not caused by the introduction of low-quality sequences in the analysis.

In the revised MS the sequences workflow is clarified as follows (added/replaced text in *italics*):

'Subsequently the TRIMMOMATIC software (Bolger et al., 2014) was used to remove low-quality sequences *starting with the following settings:* SLIDINGWINDOW:4:10 MINLEN:300 (for Bacteria); SLIDINGWINDOW:6:13 MINLEN:450 (for Archaea). In case of bacteria data this step was performed before the merging of reverse and forward reads with PEAR (Zhang et al., 2014). Merging of the archaeal reads was done *before* removing low-quality sequences in order to enhance the number of retained reads *due to* long archaeal 16S fragments. All sequences were quality controlled with FastQC (Andrews, 2010). Where necessary, more sequences were removed with TRIMMOMATIC with larger sliding window scores until the FastQC quality control was passed (average quality score per sample >34 for Bacteria and >22 for Archaea).' [Lines 157-166]

We agree that the ASV approach has a higher taxonomic resolution than OTU clustering. However, for the purpose of this paper the resolution returned by SWARM (i.e. "species" level) appears appropriate, as it allowed to distinguish microbial communities associated with nodules and sediments (as shown in Figure 2). Furthermore, according to our experience in other studies >90% of OTUs generated by SWARM overlap with variants (ASVs) identified with Dada2.

Regarding Figure 1, we do not fully agree with the Reviewers' view that the panels repeatedly show the "same information". Diversity indices and unique OTUs are reported for Bacteria and Archaea in the upper and lower panels, respectively – hence the upper and lower rows of panels refer to independent data sets. As the reviewers are certainly aware, the diversity indices presented in the first four plots or each row differ in their ecological meaning: i) total number of OTUs (H0) provides overall information about alpha-diversity, ii) exponential Shannon (H1) considers species richness and equitability, iii) inverse Simpson (H2) accounts for dominant taxa, and iv) chao1 accounts for rare taxa. Here this was calculated with the same number of sequences for each sample, thus it is not affected by sequencing depth. The last plot shows the contribution of unique OTUs to the total number of OTUs and, hence again has a different focus. While the pattern shown in the different panels may be visually similar, we are still convinced that each panel contains important information and should be presented to the reader.

Therefore, we would like keep the figure in the revised version of the MS. In our initial response to reviewers, we expressed our willingness to move plots for Archaea to the supplementary information if requested by the editor because Archaea contribute only minor to the total diversity as compared to Bacteria. There was, however, no such request so far.

RC1> A second major effort of this work is to identify taxa that are differentially abundant between nodules and sediments. While the text in Lines 244-261 describes these differences, and Table 4 includes the result of Aldex2 analysis, we don't find that Figure 4 visually conveys these differences in an easily digestible way and suggest using differential log abundance plots to more clearly show which taxa vary between the sample types.

AR> We thank the reviewers for sharing their thoughts about improving the data representation in Fig. 4. In the revised MS Figure 4 has been replaced by a fold-change plot showing genera enriched in nodules compared to those found in sediments. The original Figure 4 and Table 3 have been moved to the supplementary information (now Figures S2 and Table S3, respectively).

RC1> Another major focus of this work is the comparison of the microbial community structures between the Peru Basin nodules and those of the CCZ. I think that the paper could be improved by providing some kind of summary graphic or schematic that visually explains the differences, and their causes, as described in the text. For example, a cartoon illustrating that the lower OC

flux in the CCZ leads to nodules that look like X with communities that look like Y and perform Z functions, versus how those conditions are different at the Peru Basin. Such a summary graphic could really help simplify the presentation of the major recommendations from this work in a way that is easy to grasp, which will be especially helpful for policy makers thinking about deep-sea mining.

AR> We appreciate the suggestion of the reviewers and we generally agree. However, such a scheme would need to be supported by a deeper analysis that would require a more comprehensive dataset. Such a generalized analysis unfortunately cannot be carried out at the moment due to the limited number of studies and different sequencing methods applied to investigate microbial communities in different nodules fields. The aims of this study were to explore the role of nodules in deep-sea microbial diversity, their potential role in ecosystem functions, and a comparison of our results with data available from other nodule field regions (i.e. CCZ). Our results highlight the importance of nodules in hosting specific microbes with potentially important functions, which differ from those reported for CCZ. While the number of samples and differences in methods do not allow a generalization, we felt the need to point out important ecological questions and hypotheses that are relevant in the deep-sea mining context, but that are not yet solved and should be addressed in future studies.

This consideration is highlighted in "Conclusions" of the revised MS by adding the following statement (added/replaced text in *italics*):

'However remarkable differences in *microbial* community composition (e.g. Mn-cycling bacteria, nitrifiers) between the CCZ and the Peru Basin also show that environmental settings (*e.g.* POC flux) and features of FeMn nodules (e.g. metal content, *nodule-attached fauna*) may play a significant role in structuring the nodule microbiome. *Due to limitations in the available datasets and methodological differences in the studies existing to date, findings are not yet conclusive and cannot be generalized. However, they* indicate that microbial community structure and function would be impacted by nodule removal. *Future studies need to look at these impacts in more detail and should address regional differences*, to determine the spatial turnover and *its environmental drivers*, and the *consequences regarding* endemic types.' [Lines 485-493]

RC1> A question: in the methods, there is mention of collecting samples for cell abundance determination, but such data are not presented in this paper. Is it possible to include such data? This would help to evaluate if the "hot spot" idea discussed in the paper correlates to cell biomass - i.e. is lower diversity correlated to higher biomass?

AR> Unfortunately, we do not have cell counts for manganese nodules. Originally, we planned to include cell counts (AODC and CARD-FISH) for sediments. However, these data are reported already in another study on the impact of mining on sediment microbial communities and their biogeochemical functions which was under revision at the moment of MS submission but will be available at the time of the publication of this study (Vonnahme T.R, Molari

M., Janssen F., Wenzhöfer F., Haeckel M., Titschack J., Boetius A. Effects of a deep-sea mining experiment on seafloor microbial communities and functions after 26 years. Science Advances, in press).

We clarified this issue in the revised MS and point the reader to the Vonnahme et al. publication. [Lines 125 and Lines 133-136]

RC1> A suggestion: there is some mismatch between the 3 hypotheses posed in the introduction and the three objectives posed in the discussion section. The discussion text follows the outline of the objectives, but there is not explicit "testing" of the hypotheses proposed at the beginning of the paper, and also the discussion does exactly follow the objectives as proposed. For example, discussion section 4.3 discusses metabolisms inferred from the amplicon data, not what environmental factors structure the community, as would be assumed by how objective three is worded. We recommend bringing better alignment between the hypotheses/objectives and what the data actually address.

AR> We thank the reviewers for pointing out these inconsistencies.

The first and second hypothesis (Hp1 and Hp2) have been tested using statistical tests (as described in the Methods section). Hypotheses and "primary aims/objectives" are discussed in section 4.1 and 4.2, respectively. The "secondary aim" (previously Hp3) was addressed by deducing potential metabolic functions and habitat features/preferences from the taxa that were significantly enriched in the nodules (based on statistical testing) and what we know from descriptions of closely related organisms. From these results and comparison with CCZ microbial data we suggested potential environmental factors that can have a major role in shaping the microbial community on nodules. Based on the limited available data, however, these suggestions cannot be rigorously tested. The "secondary objective" was mainly discussed in section 4.3, but also partially addressed in the previous two sections.

In the revised MS we improved the alignment of hypotheses/aims with objectives by slightly refocusing the first and second hypothesis (Hp1 and Hp2) and by turning the third hypothesis into a secondary aim.

1) Hp1 [introduction]: 'nodules shape deep-sea microbial diversity '

Primary Objective [discussion part 4.1]: *compares* the microbes of nodule fields with microbiota of deep-sea sediments in other ecosystems in order to identify specific features of microbial diversity of nodule fields.

2) Hp2 [introduction]: 'nodules host a specific microbial community compared to the surrounding sediments'

Primary Objective [discussion part 4.2]: *elucidates* differences in diversity and in microbial community structure between sediments and nodules

3) Secondary aim [introduction]: 'Secondary aim of this study was to investigate the nodule features that may play a major role in shaping microbial community composition and microbially-mediated functions.'

Secondary objective [discussion primarily part 4.3]: *investigates potential microbially-mediated functions* and the major drivers in shaping microbial communities associated with nodules

The modified hypotheses / aims in the introduction [Lines 108-113] are now easily associated with the titles of sections 4.1, 4.2, and 4.3 of the discussion. Lines 298-304, 305 382, and 428]

RC1> minor suggestions:

RC1> L1: Title could be more descriptive of what the study discovered

AR> According to reviewers' suggestion, the title of revised MS is now:

"*The contribution of microbial communities in polymetallic nodules to the diversity of the deep-sea microbiome of the Peru Basin (4130 − 4198 meter depth)*"

RC1> L15 - consider removing "need to"

AR> We removed "need to". [Line 18]

RC1> L22 - Acidomicrobia, only one "i". To update throughout the manuscript.

AR> We are referring here to the class Acidimicrobiia within the phylum Actinobacteria in accordance with the NCBI taxonomy (https://www.ncbi.nlm.nih.gov/Taxonomy/Browser/wwwtax.cgi?id=84992)

We left this unchanged.

RC1> L78-79 - need consistency in the presentation of thousands of kilometers. In one instance, there is no punctuation; in the second instance, there is punctuation.

AR> We corrected this by removing punctuation. [Lines 82-83]

RC1> L80 - missing a decimal point in 0.2-0.6%?

AR> Reviewers are correct - we changed this accordingly. [Line 85]

RC1> L123 - were any negative DNA extraction controls included in this study, since low biomass might have been expected? If yes, please describe.

AR> Yes, we had negative controls. This is mentioned in lines 129-130 and table 2 of the submitted version of the MS. [Lines 143-145 of the current version]

RC1> L141 - Is there a reference that shows why these trimmomatic SLIDINGWINDOW parameters were used? They seem relaxed and would allow for sub-par quality reads to pass the QC step. Most workflows don't allow for sliding window PHRED scores of less than 28-30.

AR> We give detailed information in the general comments section above, and – as mentioned there – we provided additional information in the revised MS. [Lines 157-166]

RC1> L141 - recommendation to deposit your data processing pipeline to github or similar repository.

AR> The revised version of the manuscript includes information on where workflow and scripts used for sequence analysis can be found. [Lines 169-170]

RC1> L144 - There is a comparative "while" statement describing the differences between how bacterial and archaeal sequences were merged, but the way it is worded, it appears to describe the same order of operations.

AR> We agree and we corrected in the revised version of the manuscript as follows (added/replaced text in *italics*):

'In case of bacteria data this step was performed before the merging of reverse and forward reads with PEAR (Zhang et al., 2014). *Low*-quality *archaeal* sequences *were removed after merging the reads* in order to enhance the number of retained reads *due to the increase in* archaeal 16S fragment *length*.' [Lines 160-163]

RC1> L174 - Transforming count matrices using the center-log ratio requires a strategy for replacing zeros with a pseudo count because the presence of zeros produces NA values. There is no zero-replacement strategy described in this workflow. The Bray-Curtis distance cannot be computed on data matrices that contain negative numbers. A Center- log-Ratio transformed count matrix contains negative numbers. CLR transformed data is usually ordinated using the Aitchison distance metric or the Euclidean distance. I am unclear on how these analyses were performed in the way that they are described. Was the data log10(x+1) transformed? That transformation is compatible with the bray- curtis distance. The resulting ordinations looks correct, but I think the description in the methods section is inaccurate. Could the authors provide a document with the code used to perform these steps?

AR> Revier 1 is totally right. Indeed the Euclidean distance matrix was used and not Bray-Curtis (as also specified in caption of Figure 2).

In the revised MS we corrected this mistake as follows (added/replaced text in *italics*):

"Beta-diversity in samples from different substrates and from different sites was quantified by calculating an *Euclidean distance matrix* based on centred log-ratio (CLR) transformed OTU abundances *(function clr in R package compositions)* and Jaccard dissimilarity based on a presence/absence OTU table." [Lines 194-197]

RC1> L237 - these percentages are for all nodules in aggregate as an average, but does not show the variation between samples. I recommend including standard deviation plus/minus for each percentage.

AR> The percentage reported is not the average between/within groups, but it is the result of hierarchical clustering (function *hclust* in R package *vegan*) using the complete linkage method (data reported in Figure 4) and, hence, standard deviation cannot be reported. This information has beeen added to the revised MS as follows (added/replaced text in *italics*):

"The Jaccard dissimilarity coefficient was used *to perform hierarchical clustering (function hclust in R package vegan, using the complete linkage method), and the dissimilarity values for cluster nodes were used* to calculate the number of shared OTUs between*/within groups*." [Lines 200-202]

RC1> L338 - could the differences in relative percentages of archaea between this study and prior studies be due to difference in DNA extraction, primers used, or sequencing approach?

AR> Previous data for Archaea are only reported by Tully and Heidelberg (2013) and Shulse at al. (2016) in the CCZ. In the first study a modified phenol-chloroform extraction method was applied for DNA extraction, universal primers (U515/U1048; targeting the V4 region of the 16S rRNA gene) for PCR amplification and a Roche 454 Titanium platform for sequencing. Shulse and colleagues extracted DNA with the FastDNA Spin Kit for Soil (MP Biomedicals, USA), PCR amplification was carried out with universal primers (515f/805r; targeting the V4 region of the 16S rRNA gene) and sequencing with an illumina MiSeq platform. These pipelines indeed differ from those applied in our study and reported in the Methods section: DNA extraction with FastDNA Spin Kit for Soil (MP Biomedicals, USA), PCR with bacteria (341F/785R; targeting the V3-V4 region of the 16S rRNA gene) and archaea primers (349F/915R; targeting the V3-V5 region of the 16S rRNA gene), and sequencing with an illumina MiSeq platform. We agree with the reviewers that different methods applied make the comparison difficult, especially with data from Tully and Heidelberg (2013) where differences in methodology appear most pronounced. In the revised MS we limited the comparison to data from Shulse at al. (2016) because differences in methods are limited to the choice of primers, which however amplified the same hypervariable region of 16S rRNA gene (V4) reducing biases in the comparison [Lines 364-368]. This, however, does not change the overall difference in relative percentages of archaea in this study compared to previous work. We further added the statement that we cannot rule out that the slight differences in methodology between Shulse et al. (2016) and our study could be a possible explanation for the observed differences:

*We cannot rule out that the observed differences in microbial community structure partly reflect the different sets of primers used in our study and by Shulse et al. (2016). As both primer sets amplified the same hypervariable region of 16S rRNA gene (V4) we assume that biases are small enough to justify the comparison.* [Lines 368-371]

RC1> L359 - suggestion to add a clause to the end− of the sentence regarding nodules and sediments have distinct communities, stating that this observation is consistent with what has been found in earlier studies, and cite a few examples.

AR> We revised the MS accordingly (added/replaced text in *italics*):

'Analysis of community composition at OTU level shows that nodules and sediments host distinct bacterial and archaeal communities (Figure 2), *as previously reported also for CCZ (Wu et al., 2013; Tully and Heidelberg, 2013; Shulse et al., 2016; Lindh et al. 2017).*' [Lines 388-391]

RC1> L413 - "reductive"

AR> We replaced "reducers" with "reductive" [Line 440]

**Authors' Response ('AR')** to the interactive comment on "Microbial communities associated with sediments and polymetallic nodules of the Peru Basin" by Massimiliano Molari et al.

**Review by Anonymous Referee ('RC2')**

RC2> General comments to authors:
The manuscript by Molari et al. describes the microbial community structure associated with sediments and manganese nodules from 3 and 2 sites, respectively, within the Peru Basin.

The authors find that Gammaproteobacteria and Alphaproteobacteria are the dominant bacterial classes in sediments and manganese nodules while all archaeal communities investigated were dominated by Thaumarchaeota. However, sediment and nodule communities were found to differ significantly at the OTU level, as assessed by calculating Jaccard dissimilarity. The authors note differences in the nodule community composition (specifically, a lower relative abundance of Archaea, and a different nitrifier community) in their study in the Peru Basin as compared with communities in the Clarion-Clipperton Fracture Zone (CCZ), where previous work on microbial community composition of nodules has been done.

The strengths of the manuscript include the following:
i. There is a lack of studies of the prokaryotic diversity in the surface sediments and nodules of the Peru Basin, which has different environmental conditions than the relatively well-studied CCZ. ii. The molecular and bioinformatic methods are well-documented and the microbial community analysis is thorough.

Weaknesses of the manuscript include the following:
i. The lack of metadata associated with the various sites makes interpretation of the differing community structures among sites difficult.

AR> We appreciate the suggestion of the reviewer. However, the primary aim of this study was not to investigate and explain the variability of microbial community between sites, but between habitats (nodules and sediments). Only sedimentary metadata (e.g. pigments and organic carbon content, porewater profiles, and porosity) are available for sites investigated, and not for nodules, which precludes the quantitative characterization of the nodule habitat setting. Thus, sedimentary setting alone does not help to understand differences in microbial community structure and diversity that are observed between sediments and nodules. The metadata available for sediments in the study area as well as the discussion of variability of sedimentary environmental settings and microbial communities will soon be published (scheduled publication at the 29[th] of April) in "Vonnahme T.R, Molari M., Janssen F., Wenzhöfer F., Haeckel M., Titschack J., Boetius A. Effects of a deep-sea mining experiment on seafloor microbial communities and functions after 26 years. Science Advances, in press".

We point the reader to this publication in the revised version of the MS:

*"Sedimentary metadata (e.g. cell counts, pigments and organic carbon content, porewater profiles, and porosity) and a map of the study area are available in Vonnahme at al. (in press). Focusing entirely on sediments, that publication also includes a discussion of the variability of environmental settings and microbial communities."* [Lines 133-136]

RC2> Specific comments to the authors:

RC2> Major concerns:

1. Page 3 – 4. Somewhere in this discussion of the CCZ versus the Peru Basin I think it would be helpful to briefly let the reader know the state of hypothetical mining in each of these regions. In the CCZ, the ISA has entered into contracts with various contractors for exploration for polymetallic nodules. Is this the case in the Peru Basin as well?

AR> We clarified this point in the revised MS modifying the introduction as follows (added/replaced text in *italics*):

"Nodule accumulations of economic interest have been found in four geographical locations: the Clarion-Clipperton Fracture Zone (CCZ) and the Penrhyn Basin in the central north and south Pacific Ocean, respectively; the Peru Basin in the south-east Pacific; and in the center of the north*ern* Indian Ocean (Miller et al., 2018). *To our knowledge the Peru basin is the only region that does not have exploration activities and plans for mining so far.*" [Lines 60-64]

RC2> 2. Page 5, line 113. "Samples were collected at three sites…" For clarity I think the authors should explicitly state in the text that nodules were only collected at 2 of these 3 sites.

AR> The Reviewer is right and we clarified this issue in the revised MS as follows (added/replaced text in italics):

"Manganese nodules where sampled, using a TV-MUC, or a Remotely Operated Vehicle (*ROV KIEL6000, GEOMAR, Germany): one nodule at Reference West and four nodules at Reference South*." [Lines 126-128]

RC2> 3. Page 5, line 115. "… called "Reference Sites." I suggest directly listing the Reference Sites here in the text instead of making the reader consult Table 1, especially since the authors refer to Reference South later in the text. Could change to "… called "Reference Sites": Reference East, Reference West, and Reference South."

AR> We modified the revised MS according to the Reviewer's suggestion as follows (added/replaced text in italics):

"Samples were collected at three sites outside the seafloor area selected in 1989 for a long-term disturbance and recolonization experiment (DISCOL;

Thiel et al., 2001), for this reason they were called "Reference Sites": *Reference East, Reference West, and Reference South.*" [Lines 120-122]

RC2> 4. Page 5, line 116. Here a map of the Peru Basin (in addition to the Table already provided), with the study sites and DISCOL experiment sites marked, would be very helpful to the reader.

AR> An appropriate map is available in the Vonnahme et. al (in press) study mentioned above. We would suggest pointing the reader to that publication to avoid duplication. In our initial response to the reviewers we expressed our willingness to to provide a similar map also for inclusion in the revised version of the MS if requested by the editor. There was, however, no such request so far.

RC2> 5. Page 8, lines 226 – 232. "… significant differences were detected in sediment microbial community structure among the different sites… "Site" defined by geographic location and "Substrate" … explained a similar proportion of variation in bacterial community structure…" This was a bit surprising to me and this is where I think some physical/chemical/biological metadata about each site would be really helpful. If any is available, perhaps from other groups on the cruise, it would help add context to some of the observations here.

AR> A detailed environmental characterization of sites investigated and focused discussion of baseline condition (i.e. variability of environmental settings, community structure, and diversity between "Reference sites") will be soon available in the Vonnahme et al.(in press) paper mentioned above. Primary aims of this study were: i) to compare the microbes of nodules fields with the microbiome of other deep-sea sediments in order to identify specific features of microbial communities of nodule fields; ii) to elucidate differences in diversity and in microbial community structure between sediments and nodules, and their potential implications for microbially-mediated functions. Thus, we believe that in order to achieve these aims it is neither needed nor beneficial to provide and discuss sedimentary metadata. However, as mentioned above, we point out in the revised MS that this information can be found in "Vonnahme et al." in case the reader seeks a better understanding of the effect of environmental settings on microbial community structure in sediments of the sites investigated. [Lines 133-136]

RC2> 6. Page 8, lines 226 – 229. "…significant differences were detected in sediment microbial community structure … between communities associated with nodules and sediments at Reference South." I think it is important to state directly in the text that this site, Reference South, was the only site that had enough nodule sampling to allow the authors to do this analysis (at least I assume this is what occurred). Otherwise this sentence could be taken to mean that differences in community structure between nodules and sediments were also investigated at the other 2 sites, and no differences were found.

AR> We thank the Reviewer for highlighting this point.

We clarified this issue in the revised MS as follows (added/replaced text in *italics*):

"Also, significant differences were detected in sediment microbial community structure among the different sites (PERMANOVA; Bacteria: $R^2 = 0.384$; $p = 0.003$; $F_{2,8} = 1.87$; Archaea: $R^2 = 0.480$; $p = 0.013$; $F_{2,8} = 2.31$; Table S1), and between communities associated with nodules and sediment at Reference South (PERMANOVA; Bacteria: $R^2 = 0.341$; $p = 0.023$; $F_{1,6} = 2.59$; Archaea: $R^2 = 0.601$; $p = 0.029$; $F_{1,6} = 7.53$; Table S1), *which was the only site where the number of samples allowed for the test*." [Lines 249-254]

RC2> Minor issues to be addressed:

RC2> 7. Page 8, line 238. "Aphaproteobacteria" should be "Alphaproteobacteria".

AR> This has been corrected. [Line 262]

RC2> 8. Page 9, line 282. "Aphaproteobacteria" should be "Alphaproteobacteria".

AR> This has been corrected. [Line 307]

[revised manuscript text omitted]

**OTUs P/A table and Jaccard dissimilarity [A]**

| | Bacteria | | | | | | Archaea | | | | | | |
|---|---|---|---|---|---|---|---|---|---|---|---|---|---|
| Substrates | | Df | SS | MS | F | $R^2$ | P | | Df | SS | MS | F | $R^2$ | P |
| | Substrates | 1 | 0.5986 | 0.59863 | 2.7963 | 0.18899 | 0.002 | Substrates | 1 | 0.45274 | 0.45274 | 2.2661 | 0.18474 | 0.003 |
| | Residuals | 12 | 2.5689 | 0.21408 | | 0.81101 | | Residuals | 10 | 1.9979 | 0.19979 | | 0.81526 | |
| | Total | 13 | 3.1676 | | | 1 | | Total | 11 | 2.45064 | | | 1 | |
| Sites/Sediment | | Df | SS | MS | F | $R^2$ | P | | Df | SS | MS | F | $R^2$ | P |
| | Sites | 2 | 0.50624 | 0.25312 | 1.3286 | 0.30693 | 0.002 | Sites | 2 | 0.52048 | 0.26024 | 1.4829 | 0.37231 | 0.003 |
| | Residuals | 6 | 1.14312 | 0.19052 | | 0.69307 | | Residuals | 5 | 0.87749 | 0.1755 | | 0.62769 | |
| | Total | 8 | 1.64936 | | | 1 | | Total | 7 | 1.39798 | | | 1 | |
| Reference.South/Substrate | | Df | SS | MS | F | $R^2$ | P | | Df | SS | MS | F | $R^2$ | P |
| | Substrates | 1 | 0.48253 | 0.48253 | 2.2875 | 0.31389 | 0.035 | Substrates | 1 | 0.41752 | 0.41752 | 2.0157 | 0.33507 | 0.1 |
| | Residuals | 5 | 1.0547 | 0.21094 | | 0.68611 | | Residuals | 4 | 0.82856 | 0.20714 | | 0.66493 | |
| | Total | 6 | 1.53722 | | | 1 | | Total | 5 | 1.24609 | | | 1 | |
| Sites/Substrates (Strata=Site) | | Df | SS | MS | F | $R^2$ | P | | Df | SS | MS | F | $R^2$ | P |
| | Sites | 2 | 0.5954 | 0.29772 | 1.4698 | 0.18798 | 0.006 | Sites | 2 | 0.46094 | 0.23047 | 1.2347 | 0.18809 | 0.027 |
| | Sites:Substrates | 2 | 0.7492 | 0.37458 | 1.8493 | 0.23651 | 0.006 | Sites:Substrate | 2 | 0.68307 | 0.34154 | 1.8297 | 0.27873 | 0.027 |
| | Residuals | 9 | 1.823 | 0.20255 | | 0.57551 | | Residuals | 7 | 1.30663 | 0.18666 | | 0.53318 | |
| | Total | 13 | 3.1676 | | | 1 | | Total | 11 | 2.45064 | | | 1 | |

CLR: centered log-ratio; P/A: presence/absence; Df: degrees of freedom; SS: sum of the squares; F: statistic *F-ratio*; P: probability level.

[A] based on 100 sequence re-samplings per sample to the smallest dataset (40613 sequences for Bacteria and 1835 sequences for Archaea).

**Table S3.** Genera differentially abundant in nodules and sediments (ALDEx2: glm adjusted p<0.01; KW adjusted p<0.05). In bold the most abundant genera (≥0.5 %) at least two times more abundant in nodule than in sediment; in italic the genera exclusively present (i.e. unique) in nodules. Base 2 logarithm of the ratios between geometric mean centred sequences number of nodule (Nod) and sediment (Sed), and average of the sequences contribution of total number of sequences (%) retrieved in nodules and in sediments are shown.

| Enriched in Nodule | LOG2(Nod/Sed) | Nodule (%) | Sediment (%) |
|---|---|---|---|
| *Sphingomonadaceae_unclassified* | – | *0.04* | *0.00* |
| *Filomicrobium* | – | *0.01* | *0.00* |
| Geminicoccaceae_unclassified | 4 | 0.12 | 0.01 |
| Methyloceanibacter | 4 | 0.17 | 0.02 |
| Robiginitomaculum | 4 | 0.09 | 0.00 |
| Mesorhizobium | 3 | 0.25 | 0.01 |
| **Cohaesibacter** | **3** | **0.78** | **0.10** |
| OPB56_unclassified | 3 | 0.03 | 0.00 |
| 67-14_unclassified | 3 | 0.31 | 0.06 |
| Syntrophaceae_unclassified | 3 | 0.06 | 0.01 |
| Maribacter | 3 | 0.06 | 0.01 |
| **Methyloligellaceae_unclassified** | **2** | **1.46** | **0.31** |
| Entotheonellaceae_unclassified | 2 | 0.20 | 0.04 |
| Blastocatella | 2 | 0.18 | 0.04 |
| Calorithrix | 2 | 0.03 | 0.01 |
| **Hyphomicrobiaceae_unclassified** | **2** | **2.72** | **0.71** |
| Planctomicrobium | 2 | 0.05 | 0.01 |
| Simkaniaceae_unclassified | 2 | 0.13 | 0.03 |
| Microtrichaceae_unclassified | 2 | 0.13 | 0.03 |
| LD1-PA32_unclassified | 2 | 0.05 | 0.01 |
| **Subgroup 17_unclassified** | **2** | **1.03** | **0.27** |
| **JdFR-76** | **2** | **0.93** | **0.26** |
| **Subgroup 9_unclassified** | **2** | **1.26** | **0.42** |
| Chlamydiales_unclassified | 2 | 0.16 | 0.06 |
| **SAR324 clade(Marine group B)_unclassified** | **2** | **3.12** | **1.10** |
| Vermiphilaceae_unclassified | 2 | 0.14 | 0.05 |
| Acanthopleuribacter | 1 | 0.04 | 0.01 |
| Bythopirellula | 1 | 0.04 | 0.01 |
| **Nitrospina** | **1** | **3.79** | **1.72** |
| Gemmataceae_unclassified | 1 | 0.04 | 0.02 |
| Planctomycetacia_unclassified | 1 | 0.05 | 0.03 |
| SM1A02 | 1 | 0.29 | 0.15 |
| Ekhidna | 1 | 0.17 | 0.09 |
| Phycisphaeraceae_unclassified | 1 | 0.49 | 0.27 |
| **AqS1** | **1** | **1.10** | **0.66** |
| Microtrichales_unclassified | 1 | 0.19 | 0.08 |
| **Pirellulaceae_unclassified** | **1** | **1.31** | **0.75** |
| pltb-vmat-80_unclassified | 1 | 0.05 | 0.00 |
| **Pir4 lineage** | **1** | **1.60** | **0.91** |
| **Alphaproteobacteria_unclassified** | **1** | **7.15** | **4.44** |
| Babeliales_unclassified | 1 | 0.10 | 0.07 |
| Parvularculaceae_unclassified | 1 | 0.06 | 0.04 |
| PAUC43f marine benthic group_unclassified | 1 | 0.54 | 0.36 |
| Subgroup 10 | 0 | 0.75 | 0.67 |
| Aquibacter | 0 | 0.83 | 0.73 |
| Cyclobacteriaceae_unclassified | 0 | 1.76 | 1.70 |
| Gemmatimonadaceae_unclassified | 0 | 1.17 | 1.15 |
| Rhodothermaceae_unclassified | 0 | 0.38 | 0.39 |
| Total | | 35.37 | 17.82 |

| Enriched in Sediment | LOG2(Nod/Sed) | Nodule (%) | Sediment (%) |
|---|---|---|---|
| Planctomycetales_unclassified | -0.02 | 0.44 | 0.49 |
| Lutibacter | -1 | 0.00 | 0.02 |
| Chloroflexi_unclassified | -2 | 0.03 | 0.09 |
| AT-s3-28_unclassified | -2 | 0.03 | 0.09 |
| Chitinophagales_unclassified | -2 | 0.05 | 0.16 |
| Bacteriovoracaceae_unclassified | -2 | 0.04 | 0.14 |
| Nannocystaceae_unclassified | -2 | 0.02 | 0.07 |
| Cellvibrionaceae_unclassified | -2 | 0.02 | 0.08 |
| OM182 clade_unclassified | -2 | 0.13 | 0.47 |
| Candidatus Komeilibacteria_unclassified | -2 | 0.01 | 0.03 |
| Roseobacter clade NAC11-7 lineage | -2 | 0.04 | 0.11 |
| Bacteroidia_unclassified | -2 | 0.03 | 0.11 |
| IS-44 | -2 | 0.05 | 0.20 |
| Oligoflexaceae_unclassified | -2 | 0.01 | 0.08 |
| Lentimicrobiaceae_unclassified | -2 | 0.01 | 0.04 |
| Marinoscillum | -3 | 0.02 | 0.08 |
| Anaerolineaceae_unclassified | -3 | 0.05 | 0.36 |
| Colwelliaceae_unclassified | -3 | 0.01 | 0.13 |
| Subgroup 7_unclassified | -3 | 0.00 | 0.03 |
| Peredibacter | -3 | 0.01 | 0.05 |
| Marinimicrobia (SAR406 clade)_unclassified | -4 | 0.01 | 0.07 |
| Total | | 1.00 | 2.91 |

---

## Author Response (AR2)

**Authors' Response ('AR')** to the Associate Editor comment on "The contribution of microbial communities in polymetallic nodules to the diversity of the deep-sea microbiome of the Peru Basin (4130 − 4198 meter depth)" by Massimiliano Molari et al.

**Associate Editor Dr. Denise Akob ('AE') Comments**

AE> I believe this should be the GenBank Database and please provide a reference for both BLASTn and GenBank

AR> we corrected the text and add the following reference for BLATS: "Altschul, S. F., Gish, W., Miller, W., Myers, E.W. and Lipman, D. J.: Basic local alignment search tool. J. Mol. Biol. 215:403–410, 1990." [Lines 172-175 and 515-516]

AE> change to OTU

AR> Thanks. We corrected this. [Line 215]

AE> change to OTUs

AR> Thanks. We corrected this. [Line 216]

AE> do you mean Gemmatimonadetes?

AR> yes. We thank Editor for highlighting this typo. We corrected with "Gemmatimonadetes" [Line 302].

AE> I suggest changing "no cultivates" to "no cultured relatives"

AR> We thank the Editor for the suggestion and we modified the text accordingly. [Line 314]

AE> I don't agree with the use of the word turnover as you didn't look at changes over time. I suggest changing to shift or differ.

AR> We thank the Editor for suggestion. However we believe that "turnover" here is appropriate to describe spatial shift in OUTs composition between microbial communities, and compared to beta-diversity values reported in literature. In order to clarify that we refer to spatial and NOT temporal turnover, we reformulate the sentence as follow:
"Beta-diversity of microbial community structure in the Peru Basin sediments showed remarkable spatial OTU turnover already on a local scale (<60 km; Figure S2), which is at the higher end of previous microbial beta-diversity estimates for bathyal and abyssal seafloor assemblages (Jacob et al., 2013; Ruff et al., 2015; Bienhold et al., 2016; Walsh et al., 2016; Varliero et al., 2019)." [Lines 372-375]